



# Cryptotephra in the East Antarctic Mount Brown South ice core

Margaret Harlan[1,2,3], Jodi Fox[3,4], Helle Astrid Kjær[2], Tessa Vance[1], Anders Svensson[2], and Eliza Cook[2]

[1]Australian Antarctic Program Partnership, Institute for Marine and Antarctic Studies, University of Tasmania, Hobart, Australia
[2]Physics of Ice, Climate, and Earth, Niels Bohr Insitute, University of Copenhagen, Denmark
[3]Institute for Marine and Antarctic Studies, University of Tasmania, Hobart, Australia
[4]Department of Geology and Paleontology, National Museum of Nature and Science, Tsukuba, Japan

**Correspondence:** Margaret Harlan (margaret.harlan@utas.edu.au)

**Abstract.**

Ice cores contain stratified layers of impurities scavenged from the atmosphere, which are a vital tool for investigating the Earth system. Reconstructing past eruption records by way of ice core tephrochronology can help us understand ash dispersal, atmospheric circulation processes, and the impacts of volcanic eruptions on climate. This study presents the coastal East

Antarctic Mount Brown South (MBS, 69.11°S, 86.31°E; 2084 m ASL) ice core as an untapped tephrochronological archive. We utilize a novel cryptotephra sampling plan, integrating ice core data, HYSPLIT air parcel trajectories, and known eruption records, and identify two distinct cryptotephra horizons at ~13.3 and ~17.9 m depth in the MBS-Alpha ice core. We also find sparse tephra grains throughout the core. Through geochemical characterization with electron probe microanalysis (EPMA), we correlate the two cryptotephra horizons with the 1991 eruption of Cerro Hudson and the continuous eruptions of Mt. Erebus

throughout the mid-1980s. The volcanic horizons identified here underscore the role of MBS in extending the regional volcanic record, helping to constrain ice core dating efforts, and enhancing understanding of volcanic ash dispersal to East Antarctica.

## 1 Introduction

Ice core records are important archives of the changing earth systems. They record recent climate events in detail, including volcanic eruption histories, and in some cases the strength and source of such eruptions (Svensson et al., 2020; Sigl et al., 2014;

Lowe, 2011; Abbott et al., 2024; Narcisi et al., 2012; Kurbatov et al., 2006; Dunbar et al., 2003). Ice sheets and ice caps can contain both soluble (e.g. volcanic sulfate) and insoluble (e.g. ash) volcanic products, deposited on the ice, providing valuable information on past volcanic eruption histories, and analyzing these products in ice cores can deepen our understanding of volcanic climate forcing, and help to refine ice core chronologies (Gao et al., 2008; Castellano et al., 2004; Sigl et al., 2014, 2015; Lohmann and Svensson, 2022; Lin et al., 2022).

Tephrochronology, the use of tephra (material ejected during explosive volcanic eruptions) as isochronous horizons in stratigraphic archives such as ice cores and sediment records can be used to date archives, synchronize geographically distinct records, and inform about past volcanic events (Lowe, 2011; Geyer et al., 2023; Cook et al., 2022). While volcanic horizons can be identified by soluble tracers like volcanic sulfate or acidity, characterization and geochemical fingerprinting of tephra



can be used to identify the specific volcanic sources for eruptions events (Cook et al., 2018; Lin et al., 2022; Svensson et al.,
2020; Geyer et al., 2023; Lowe, 2011).

Ice core records are often geographically distal or ultra-distal archives of volcanic ash, and the tephra found are typically
micrometer-scale volcanic glass shards from volcanic ash fallout transported long distances by atmospheric circulation pro-
cesses (Lowe, 2011; Geyer et al., 2023). Locating and identifying cryptotephra (tephra not visible to the naked eye due to size
or sparseness) in ice cores is especially difficult, as volcanic markers such as sulfate, conductivity, or acidity do not always
co-occur with tephra deposits from the same eruption, and comprehensive sampling must be undertaken to produce a full
tephrochronological framework (Cook et al., 2018; Lin et al., 2022; Abbott et al., 2024; Lowe, 2011; Narcisi et al., 2010; Cook
et al., 2022; Basile et al., 2001).

Tephra previously identified in other studies of Antarctic ice cores come from local and regional sources, including Antarctic
(Narcisi et al., 2006; Basile et al., 2001; Abbott and Davies, 2012) and sub-Antarctic island volcanoes (Basile et al., 2001;
Abbott et al., 2024; Narcisi et al., 2012). Additionally, tephra found in Antarctic ice and snow has been correlated to large
eruptions from ultra-distal eruptions e.g. Aotearoa/New Zealand (Taupo; Dunbar et al. (2017)), South America (Cerro Hudson,
Puyehue-Cordón Caulle; Abbott et al. (2024); Narcisi et al. (2012); Koffman et al. (2017)), and possibly even Mexico (El
Chichón; Palais et al. (1992)).

Many efforts in recent years have been undertaken to improve Antarctic tephrochronologies, using ice cores from across
the continent including Talos Dome (Narcisi et al., 2012), Siple Dome (Kurbatov et al., 2006), Vostok (Basile et al., 2001;
Narcisi et al., 2010), and others (Abbott et al., 2024; Narcisi et al., 2005, 2010). Ice cores spanning millenia can be used to
produce tephrochronologies spanning the Holocene or Last Glacial period, often with an aim to constrain timings of changes to
global climate (Lin et al., 2022; Lohmann and Svensson, 2022; Cook et al., 2022; Castellano et al., 2004; Abbott and Davies,
2012). Short ice cores from higher resolution sites, on the other hand, can provide important means for investigating volcanic
eruptions and climate in detail in the more recent past (Abbott et al., 2024; Plunkett et al., 2023; Piva et al., 2023; Narcisi and
Petit, 2021; Sigl et al., 2014, 2013; Narcisi et al., 2012; Gao et al., 2008).

A number of volcanoes with the potential to transport volcanic material to Antarctica have been active during the satellite era
(1978 to present) (Global Volcanism Program, 2024). Volcanic products most commonly seen in Antarctic records include those
from Antarctic (e.g. Mt. Erebus) and Sub-Antarctic Island volcanoes (including South Sandwich and South Shetland Islands),
as well as lower latitude volcanoes in Chile and Aoteroa/New Zealand (Narcisi et al., 2012; Dunbar et al., 2017; Koffman et al.,
2017). Such satellite era eruption events are typically well observed through comprehensive monitoring programs and satellite
remote sensing (Global Volcanism Program, 2024; Francis et al., 1996; Poland et al., 2020).

The Mount Brown South (MBS) ice cores, comprising comprising a deep ice core (295 m) and 3 surface cores (~20 - 25 m),
were drilled in coastal East Antarctica during the 2017-2018 austral summer field season (69.11°S, 86.31°E, 2084 m ASL; Fig.
1). The MBS cores provide a new, high resolution climate archive spanning 1137 years, with shallow cores providing duplicate
records from the satellite era to 2017 (Vance et al., 2024a).

The MBS site was selected for its teleconnections and strong climatological link to the Southern Indian Ocean, providing
a millenial-length past climate record for a region underrepresented in the existing array of Antarctic ice core records (Vance





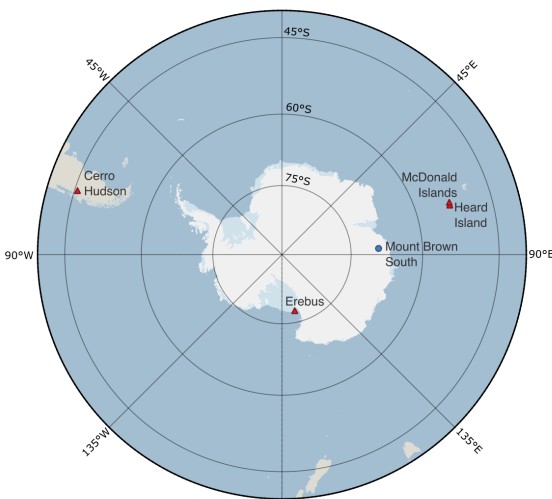

**Figure 1.** Map of Antarctica showing the location of the MBS ice core site, as well as volcanic regions of interest relevant to this study (basemap provided by the SCAR Antarctic Digital Database, accessed via the Norwegian Polar Institute's Quantarctica package (Matsuoka et al., 2018)).

et al., 2016, 2024a). Subsequent analyses have shown that East Antarctic ice core records (from MBS as well as the Law Dome
ice core) provide insight into past climate conditions, including capturing anthropogenic changes to atmospheric greenhouse
gases (Etheridge et al., 1996), El Niño-Southern Oscillation (Crockart et al., 2021) and Australian hydroclimate and bushfire
conditions (Udy et al., 2022, 2024), as well as regional events including atmospheric rivers and extreme precipitation events
(Jackson et al., 2023; Gkinis et al., 2024b; Zhang et al., 2023).

Due to the coastal Antarctic site, as well as the known teleconnections across the region, we propose that MBS is well situated
amongst Antarctic ice cores to receive and store wind-blown volcanic ash, making it a useful tephrochronological archive. We
present an investigation of cryptotephra in the MBS ice cores, in order to assess potential preservation of volcanic ash, and
better characterize the tephra transport pathways in the region. This study focuses on the satellite era record from MBS (1978
to present) as a first investigation of cryptotephra in MBS. Our characterization of the cryptotephra horizons together with
atmospheric modelling and satellite derived data will inform future design and interpretation of tephrochronology paleo-record
studies of MBS and ice cores across Antarctica.

## 2   Materials and methods

### 2.1   Mount Brown South ice cores

The MBS-Main core (4.25-294.785 m depth) is supported by three surface firn cores: MBS-Alpha (surface to 20.41 m), -Bravo
(surface to 20.225 m), and -Charlie (surface to 25.89 m). Mean accumulation derived by annual layer counting throughout the





satellite era is 0.298±0.07 m yr$^{-1}$ IE for the main core and 0.309±0.08 m yr$^{-1}$ IE for the MBS-Alpha core (Crockart et al.,
2021). This correlates with the ERA5 estimated site annual accumulation of 0.302±0.05 m yr$^{-1}$ IE (Crockart et al., 2021).
The climatology and site conditions of Mount Brown South are discussed in detail in Vance et al. (2016, 2024a), Crockart et al.
(2021) and Jackson et al. (2023).

The MBS-Main and MBS-Alpha cores have been analyzed for both trace impurities and water isotopes ($\delta^{18}$O) (Gkinis et al.,
2024a, b; Jackson et al., 2023). Measurements were performed on discrete samples at 3 cm resolution, using ion chromatogra-
phy (chemistry/trace impurities, Vance et al. (2024b); Harlan et al. (2024b)) and cavity ring down spectroscopy (water isotopes,
Moy et al. (2024); Gkinis et al. (2024b)) at the Institute for Marine and Antarctic Studies at the University of Tasmania and
Physics of Ice, Climate and Earth, Niels Bohr Institute, at the University of Copenhagen. Additionally, the MBS-Main core was
analyzed using continuous flow analysis for trace impurities (Harlan et al., 2024b) and water isotopes (Gkinis et al., 2024b),
also at the University of Copenhagen.

## 2.2 Ice core sampling

Sample depths were selected in order to target volcanic material entrained in mid-latitude moisture from the southern Indian
Ocean, which is the origin of significant snowfall accumulation to MBS (Jackson et al., 2023; Udy et al., 2021; Vance et al.,
2024a). This could include material from volcanoes in the Kerguelen Plateau region (Heard and McDonald Islands) with the
aim to better understand volcanic fertilization over the Kerguelen Plateau. In order to refine our sampling strategy, we targeted
our sampling toward air masses that passed over the Heard and MacDonald Islands region prior to arriving at MBS (based on
HYSPLIT back trajectory modeling - section 2.5). This forms part of a broader project investigating iron fertilization in the
Southern Ocean, however, for the purposes of this study, we consider tephra layers of any origin. Sampling was conducted
from the MBS-Main and MBS-Alpha cores (Fig. 2) depth ranges covering the satellite era (down to ∼20 meters).

As this is the first investigation of cryptotephra in the MBS ice cores, for this study we focus on the satellite era (1978-
present). This allows us to utilize atmospheric circulation modeling to assess volcanic ash transport, and to consider satellite
volcanic observations to inform our volcanic matching efforts. Additionally, knowledge of the magnitude and timing of eruption
events identified in satellite era ice cores can help tune future efforts to reconstruct events identified from volcanic horizons
in the more distant past. Furthermore, from a practical perspective, the duplicate cores covering this period allow for larger
sample volumes for analysis.

First, a broad low resolution screening of target depths was conducted from the MBS Main core based on the atmospheric
modelling results and using samples ranging from 15-18 cm in length (six samples per one-meter-long ice core segment, Fig.
2b, measured to the nearest half-centimeter), and cut with a band saw from the outer edge of the core (∼2 cm$^2$ cross sectional
area). With a minimum annual layer thickness throughout the satellite era of 0.256 m firn depth (mean 0.501 ±0.137 m, based
on MBS2023 (Vance et al., 2024b)), this sample size represents at least sub-annual sampling resolution. MBS Main core
samples were melted in polycarbonate bottles and centrifuged (5 minutes, 3000 rpm). Resulting concentrate was evaporated on
frosted glass slides and sample material was coated in low-viscosity epoxy resin (Logitech type 301 2-part epoxy resin). For
efficiency, these samples were investigated using optical microscopy and prioritized based on presence or absence of potential



**Figure 2.** a. Schematic describing two-phase sampling strategy. A first low-resolution set of "screening samples" were prepared from the MBS main core, followed by a resampling from the MBS-Alpha core, due to larger available sample volume. Samples from the MBS-Alpha core were assessed based on visual examination using a petrographic microscope, SEM-EDS with automated mineralogy (AMICS), BSE imaging, and finally geochemistry was measured using EPMA-WDS. Examples shown of sampling procedure for selected samples from 13 - 14 m depth from the MBS-Main (b.) and MBS-Alpha (c.) cores.

tephra grains. Out of a total of 65 exploratory samples from the MBS Main core, 22 samples were selected based on visual inspection using optical microscopy for re-sampling of the corresponding depth ranges in the MBS-Alpha core (Fig. 2a. 1-2).

Second, a higher resolution sampling was conducted from the MBS-Alpha core (due to larger availability of sample material remaining; Fig. 2c). The depths from the 22 selected MBS Main core samples were transposed to the corresponding MBS-Alpha core depths (based on the annual horizons presented in Vance et al. (2024a), Fig. 2 a. 3), resulting in 70 sample depths ranging from 4-8 cm in length (Fig. 2a. 5). Samples were measured to the nearest half-centimeter and cut using a thin bladed



pull-saw. After ∼1 mm of material from each of the outer edges was removed using a ceramic blade, the samples had a cross sectional area of ∼10 cm$^2$. Samples were melted at ambient temperature in rinsed sterile Whirl-Pak bags and centrifuged (5 minutes, 3000 rpm) in acid-washed 15 ml centrifuge tubes. Resulting concentrate was pipetted onto polyimide (Kapton) tape adhered to flat glass plates, set to evaporate on a hot plate at 60 °C and backfilled within 25 mm acrylic rings with low-viscosity epoxy (Struers EpoFix) to create round resin mounts.

The sample mount surfaces were polished using 1 micron aluminum oxide polishing compound to remove surface resin and expose any tephra grains present in the mounts, then cleaned in an ultrasonic bath to remove excess polishing compound. Samples were examined using transmitted and reflected light using a petrographic microscope, and mounts identified as containing tephra grains of suitable size and quality for future EPMA (e.g. with large enough surface area for the EPMA beam accounting for mineral inclusions and/or vesicles; Fig. 2a. 6) were carbon-coated in preparation for backscatter electron (BSE) imaging,

energy dispersive spectroscopy (SEM-EDS, FEI MLA 650 ESEM), and electron probe microanalysis (EPMA; 2a. 7-9).

### 2.3 Sub-annual age determination

The MBS Main and surface cores have been dated using independent layer counting and volcanic matching (Vance et al., 2024a). The MBS2023 chronology provides depths for annual horizons for the main and surface cores (Vance et al., 2024b). As the focus of this study is on satellite-era tephrochronology, and the annual layer thickness is suitable for sub-annual sampling,

we are able to provide much higher resolution samples, when compared to many longer-term tephrochronologies of Antarctic and Greenlandic deep ice cores (e.g. Cook et al. (2022); Narcisi et al. (2005, 2012)).

Due to the high resolution of our samples, it is helpful for source identification to be able to provide sub-annual dates for our sample depths, however a sub-annual chronology has not been published for the MBS cores. MBS is characterized by variability in accumulation (both between years and within years; Vance et al. (2024a); Crockart et al. (2021); Jackson et al. (2023))

associated with extreme precipitation events at the site (Jackson et al., 2023). This variability limits the accuracy of a sub-annual age scale derived by linear interpolation between annual horizons. Assessment of sub-annual dating for particular tephra sample depths relies on subjective interpretation of the relative position of the sample depths in comparison with seasonally-varying chemical species measured in the ice core and/or water isotope measurements (Vance et al., 2024b; Moy et al., 2024). In an effort to estimate sub-annual ages for the samples in this study, we use variations in austral summer peaking sulfate-on-

chloride (SO$_4^{2-}$/Cl$^-$) and $\delta^{18}$O and austral winter peaking sodium (Na$^+$) to approximate the time of year to which the samples may correspond.

### 2.4 Geochemical analysis of tephra

The mounted volcanic glass shards were analyzed using EPMA at the Central Science Laboratory (CSL) at the University of Tasmania (Fig. 2) in 2023 and early 2024. Analyses were conducted on a JEOL JXA-8530F Plus field emission microprobe

with 5 wavelength dispersive spectrometers. Single point analyses used a 2 $\mu$m beam diameter with 4 nA beam current, 15 kV accelerating voltage. Analytical conditions were chosen to balance minimizing beam damage, particularly Na ion migration, while maintaining a beam size suitable for our smallest (≤5 μm) shards. Concentrations of 14 major and minor elements were





measured in the individual shards. A series of mineral standards were used for instrument calibration, a rhyolitic glass secondary standard (VG-586, NMNH 72854) was used to validate accuracy and precision of measurements between measurement
sessions. See Supplementary information for full details on the EPMA measurements and standards.

Data was detection limit filtered (99% confidence), and erroneous analyses of non-volcanic material (e.g. quartz grains or other mineral particles in the samples) were removed from the dataset. The majority of the analyses have analytical total oxide values of >90%, however due to the very small size of the shards (<5μm), some analyses resulted in low analytical totals. Iverson et al. (2017) show with their "broad beam overlap" method that analytical totals as low as 67% are statistically
similar to the same analyses with higher analytical totals, with only a slight decrease in precision. Narcisi et al. (2019) show reliable results with analytical totals as low as 60%. Following this approach, analyses with analytical totals of at least 50% are presented here, however the few data with totals below 60% are considered with some caution. Data was corrected for Cl concentrations and normalized to 100% anhydrous based on the as recommended by Iverson et al. (2017). Full geochemical results and secondary standards measurements can be found in the supplementary materials.

## 2.5 Atmospheric circulation modeling

The Hybrid Single Particle Lagrangian Integrated Trajectory model (HYSPLIT; Stein et al. (2015)) was used to investigate potential sources of volcanic glass shards identified at the MBS site. We generated six-hourly ten day back trajectories originating from 1500 m AGL at the MBS site (Fig. 1), during the time periods associated with abundant tephra shards (>4 shards in one sample). For ease of trajectory generation and repeatability, we used the PySPLIT package (Warner, 2018). NCEP/NCAR
reanalysis data (Kalnay et al., 1996) was used for the meteorology conditions. Individual trajectories were clustered using the built-in clustering algorithm in HYSPLIT. The aim of the algorithm is to identify a number (user-defined) of maximally distinct clusters while minimizing variability within each cluster (Stein et al., 2015).

## 3 Results

Glass shards were identified using a petrographic microscope in 48 out of the 70 samples prepared from the MBS-Alpha core.
Those 48 samples were inspected using backscatter electron imaging and SEM-EDS with automated mineralogy, and 29 were selected for further analysis using EPMA. 12 of those 29 samples contained glass shards large enough for reliable geochemical measurements using EPMA (Fig. 2). A summary of all glass shards identified in the MBS-Alpha core is reported in Table 1.

### 3.1 Tephra morphology

Glass shards are most abundant in the 13.28-13.34 and 16.87-16.915 m depth samples. The 13.28-13.34 m sample has the
largest glass shards (Fig. 3), ranging from 10-20 microns along the longest axis. All shards are angular, while the larger shards in this sample have cuspate, bubble-wall, or y-junction morphologies and smaller shards appear more blocky or platy. Glass shards in the 16.87-16.915 m sample are typically smaller, <10 microns, and only one shard ~15 microns across the longest





**Table 1.** Summary of samples containing volcanic glass shards. Sample depths were measured to the nearest half-centimeter from the top of each ice core segment during sampling in the freezer laboratory and assimilated with core top depths; measurement error associated with core length estimates are in line with information provided in Vance et al. (2024a). Ages provided here are approximated based on MBS2023 chronology (Vance et al., 2024a). Composition classifications are based on total alkalis vs silica (TAS) diagram (Le Bas et al., 1986), the number of shards of each composition is given in parentheses. Further discussion of proposed volcanic sources (bold text in table) can be found in section 4.

| Sample ID | Sample depth range | Approx. date | Total shard count | Composition *(Le Bas et al., 1986)* | Proposed source |
|-----------|-------------------|--------------|-------------------|-------------------------------------|-----------------|
| 7-6 | 6.39-6.45 m | Mid 2004 | 3 | Phonolite (1), Dacite (2) | - |
| 7-7 | 6.45-6.51 m | Mid 2004 | 1 | Rhyolite (1) | - |
| 8-5 | 7.98-8.05 m | Mid 2001 | 3 | Rhyolite (1) | - |
| 9-10 | 8.94-9.00 m | Early 2000 | 3 | Rhyolite (1), Dacite (1), Basaltic Andesite (1) | - |
| 13-3 | 12.79-12.85 m | Mid 1992 | 1 | Rhyolite (1) | - |
| **14-1** | **13.28-13.34 m** | **Mid 1991** | **13** | **Dacite (12), Andesite (1)** | **Cerro Hudson** |
| 16-4 | 15.93-16.00 m | Early 1987 | 2 | Rhyolite (2) | - |
| 17-1 | 16.00-16.045 m | Early 1987 | 1 | Rhyolite (2) | - |
| 17-5 | 16.20-16.25 m | Mid 1986 | 1 | Trachyte/Trachydacite (1) | - |
| **17-9** | **16.87-16.915 m** | **Mid 1985** | **10** | **Phonolite (10)** | **Mt. Erebus** |
| 18-1 | 17.00-17.05 m | Mid 1985 | 1 | Trachyandesite (1) | - |
| 18-5 | 17.24-17.30 m | Early 1985 | 3 | Trachyandesite (1), Trachyte/Trachydacite (1), Rhyolite (1) | - |

axis. These shards have simpler angular morphology, and include blocky and platy shapes, some have fluted or cuspate edges. The remaining samples contain glass shards ranging from <5 to 15 microns, with mostly simple angular morphologies.

## 3.2 Geochemical composition

Major element oxides of all tephra identified in the MBS-Alpha core are provided in Table 2. The MBS-Alpha glass shards have $SiO_2$ values ranging from 51.70 to 77.31 wt.%, with 1.38-9.12 wt.% $Na_2O$, and 0.28-6.0 wt.% $K_2O$ (Fig. 4, Table 2). When plotted on the total alkali-silica diagram (TAS diagram; Le Bas et al. (1986)), compositions for the glass shards include basaltic andesite, trachyandesite, andesite, dacite, trachyte/trachydacite, rhyolite, and phonolite. The most abundant compositions are phonolite (11 shards), dacite (15 shards) and rhyolite (10 shards). Remaining compositions have fewer shards (one basaltic andesite, two trachyandesite, two andesite, and two trachyte/trachydacite shards).

The glass shards in the 13.28-13.34 m sample (table 1) are mainly dacite in composition, with one andesite shard (Fig. 4). The analyses from this depth have $SiO_2$ ranging from 61.87 to 66.23 wt.%, with total alkalis ($Na_2O+K_2O$) ranging from 3.16





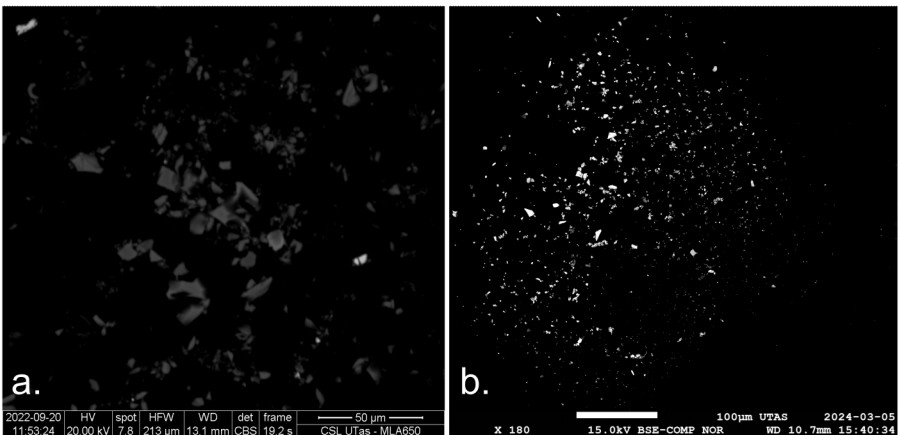

**Figure 3.** Example backscatter electron images of sample material found in the MBS-Alpha core in the 13.28-13.34 m (a) and 16.87-16.915 m (b) samples.

to 7.27 wt.%. Bivariate diagrams demonstrate there is little variation in the abundance of other major element oxides when plotted against $SiO_2$ (Figure 4).

All glass shards analyzed in the 16.87-16.915 m sample (table 1 have a homogeneous phonolitic composition. Compared to the 13.28-13.34 m horizon, these glass shards have a somewhat lower $SiO_2$ concentration, ranging from 56.31 to 58.76 wt.% and are strongly alkaline ($Na_2O+K_2O$), ranging from 12.80 to 14.23 wt.% (Fig. 4). As a group, these shards have among the highest $Na_2O$ and $K_2O$ contents of all MBS-Alpha glass analyzed here (7.04-9.12 and 4.78-5.76 wt.%, respectively).

### 3.3 Sample age estimates

The 13.28-13.34 m sample falls between the 1991 and 1992 annual horizons (13.761 and 13.099 m respectively, MBS2023; Vance et al. (2024a)). In Fig. 5, sample depth ranges are shown alongside seasonally varying isotope and chemistry species measured in the MBS-Alpha core. It is notable that during the development of the MBS2023 chronology, it was determined that trace chemistry was as reliable as $\delta^{18}O$, and at times more so, in determing annual horizon positions in the MBS records (section 4.3 in Vance et al. (2024a)). It can be seen in Fig. 5 that the 13.28-13.34 m sample falls towards the end of the austral winter sodium peak, but just at the start of the austral summer peak in sulfate-on-chloride, indicating that the glass shards were likely deposited with snow that fell in mid to late 1991. It can also be seen that the sample falls just at the beginning of a significant peak in non-seasalt sulfate, likely volcanic in origin.





**Table 2.** Major element oxide concentrations of glass tephra shards identified in the MBS-Alpha core. Colored symbols correspond with the legend in Figure 4. Analyses were conducted on a JEOL JXA-8530F Plus field emission microprobe with five WDS spectrometers. Single point analyses used a 2 $\mu$m beam diameter with 4 nA beam current, 15 kV accelerating voltage. Data are normalized to 100% using the Cl-adjusted analytical total, following the "broad beam overlap" method introduced in Iverson et al. (2017). "n.d." indicates measured values below instrument detection limit.

| Sample | | $SiO_2$ | $TiO_2$ | $Al_2O_3$ | FeO | MnO | MgO | CaO | $Na_2O$ | $K_2O$ | $P_2O_5$ | Total |
| | | wt. % | wt. % | wt. % | wt. % | wt. % | wt. % | wt. % | wt. % | wt. % | wt. % | % |
|---|---|---|---|---|---|---|---|---|---|---|---|---|
| 7-6_002* | ✕ | 66.16 | 0.09 | 20.07 | 1.00 | n.d. | 0.14 | 4.94 | 6.53 | 0.69 | 0.11 | 100.00 |
| 7-6_013 | ✕ | 65.13 | 1.63 | 14.78 | 8.61 | n.d. | 0.47 | 2.55 | 5.44 | 0.98 | 0.32 | 100.00 |
| 7-6_014 | ✕ | 59.08 | 0.99 | 19.51 | 4.27 | 0.23 | 0.81 | 1.34 | 7.61 | 5.50 | 0.29 | 100.00 |
| 7-7_006 | + | 77.16 | 0.30 | 13.11 | 0.22 | n.d. | 0.17 | 0.94 | 2.06 | 6.00 | 0.00 | 100.00 |
| 8.5_005 | ● | 77.31 | 0.13 | 13.11 | 0.40 | 0.06 | 0.07 | 0.51 | 3.96 | 4.49 | 0.03 | 100.00 |
| 8.5_007* | ● | 77.10 | 0.04 | 13.18 | 0.43 | n.d. | 0.00 | 0.48 | 4.36 | 4.36 | 0.05 | 100.00 |
| 8.5_010 | ● | 73.97 | 0.34 | 14.60 | 1.43 | n.d. | 0.43 | 1.53 | 3.22 | 4.51 | 0.06 | 100.00 |
| 9-10_014 | ▲ | 56.70 | 1.12 | 21.70 | 7.56 | n.d. | 2.20 | 5.13 | 2.58 | 2.53 | 0.43 | 100.00 |
| 9-10_017 | ▲ | 75.26 | 0.08 | 15.38 | 0.47 | n.d. | 0.40 | 0.19 | 6.71 | 1.47 | 0.00 | 100.00 |
| 9-10_021 | ▲ | 67.32 | 0.39 | 12.13 | 8.00 | 0.17 | 6.83 | 1.47 | 1.44 | 2.08 | 0.15 | 100.00 |
| 13-3_011 | ▼ | 69.48 | 0.59 | 15.77 | 2.65 | n.d. | 0.85 | 2.64 | 3.43 | 4.29 | 0.13 | 100.00 |
| 14-1_003a | ✦ | 65.80 | 1.27 | 17.16 | 5.05 | 0.19 | 1.53 | 3.24 | 2.73 | 2.71 | 0.31 | 100.00 |
| 14-1_006 | ✦ | 61.87 | 1.38 | 17.31 | 6.19 | 0.14 | 2.44 | 4.48 | 3.42 | 1.99 | 0.61 | 100.00 |
| 14-1_010a | ✦ | 64.91 | 1.29 | 16.59 | 5.45 | 0.17 | 1.82 | 3.30 | 3.46 | 2.60 | 0.35 | 100.00 |
| 14-1_011a | ✦ | 65.35 | 1.31 | 17.10 | 5.38 | 0.20 | 1.47 | 3.32 | 2.90 | 2.58 | 0.38 | 100.00 |
| 14-1_013a | ✦ | 67.79 | 1.12 | 17.27 | 4.67 | 0.16 | 1.39 | 2.77 | 1.74 | 2.83 | 0.32 | 100.00 |
| 14-1_015 | ✦ | 65.23 | 1.25 | 16.73 | 5.15 | 0.15 | 1.59 | 3.25 | 3.67 | 2.57 | 0.36 | 100.00 |
| 14-1_016a | ✦ | 66.05 | 1.17 | 17.16 | 5.47 | 0.18 | 1.47 | 3.39 | 2.26 | 2.55 | 0.36 | 100.00 |
| 14-1_017a | ✦ | 63.43 | 1.60 | 16.77 | 6.52 | 0.21 | 2.34 | 4.67 | 1.38 | 2.33 | 0.61 | 100.00 |
| 14-1_019 | ✦ | 65.57 | 1.19 | 17.00 | 5.26 | 0.20 | 1.58 | 3.23 | 2.61 | 2.76 | 0.42 | 100.00 |
| 14-1_021a | ✦ | 66.23 | 1.31 | 16.72 | 5.21 | 0.19 | 1.65 | 3.45 | 2.28 | 2.63 | 0.40 | 100.00 |
| 14-1_022a | ✦ | 65.15 | 1.30 | 16.86 | 5.28 | 0.21 | 1.70 | 3.38 | 2.92 | 2.66 | 0.50 | 100.00 |
| 14-1_024a | ✦ | 63.76 | 1.32 | 16.69 | 5.52 | 0.21 | 1.69 | 3.41 | 4.28 | 2.60 | 0.49 | 100.00 |
| 14-1_025 | ✦ | 64.48 | 1.20 | 16.75 | 5.60 | 0.19 | 1.59 | 3.34 | 3.82 | 2.56 | 0.34 | 100.00 |
| 16-4_014 | ▶ | 76.90 | 0.06 | 13.77 | 0.42 | 0.10 | 0.07 | 0.48 | 3.75 | 4.36 | n.d. | 100.00 |
| 16-4_024 | ▶ | 73.11 | 0.03 | 16.05 | 2.50 | 0.04 | 0.02 | 0.68 | 3.22 | 4.34 | n.d. | 100.00 |
| 17-1_014 | ◀ | 75.83 | 0.13 | 18.81 | 0.72 | n.d. | 0.33 | 0.67 | 3.06 | 0.28 | 0.02 | 100.00 |
| 17-5_081* | ■ | 61.27 | 0.14 | 25.36 | 1.26 | n.d. | n.d. | 1.78 | 6.25 | 3.92 | 0.00 | 100.00 |





*continued from previous page*

| Sample | | SiO$_2$ | TiO$_2$ | Al$_2$O$_3$ | FeO | MnO | MgO | CaO | Na$_2$O | K$_2$O | P$_2$O$_5$ | Total |
|---|---|---|---|---|---|---|---|---|---|---|---|---|
| | | wt. % | wt. % | wt. % | wt. % | wt. % | wt. % | wt. % | wt. % | wt. % | wt. % | % |
| 17-9 _005 | ★ | 57.02 | 1.01 | 20.14 | 5.04 | 0.27 | 0.99 | 1.91 | 7.88 | 5.22 | 0.26 | 100.00 |
| 17-9 _008 | ★ | 58.76 | 0.97 | 19.97 | 3.99 | 0.13 | 0.80 | 1.00 | 8.87 | 5.11 | 0.29 | 100.00 |
| 17-9 _009* | ★ | 57.10 | 0.98 | 19.80 | 5.81 | 0.20 | 0.97 | 1.80 | 7.04 | 5.76 | 0.31 | 100.00 |
| 17-9 _010 | ★ | 57.32 | 1.03 | 19.68 | 5.30 | 0.29 | 0.86 | 1.92 | 7.69 | 5.49 | 0.32 | 100.00 |
| 17-9 _011 | ★ | 56.48 | 0.98 | 20.16 | 4.73 | 0.31 | 0.93 | 1.76 | 9.12 | 5.11 | 0.25 | 100.00 |
| 17-9 _013 | ★ | 56.44 | 1.09 | 20.06 | 5.64 | 0.33 | 0.92 | 1.85 | 8.10 | 5.19 | 0.29 | 100.00 |
| 17-9 _018 | ★ | 57.38 | 0.90 | 20.12 | 5.04 | 0.20 | 0.96 | 1.68 | 8.56 | 4.87 | 0.18 | 100.00 |
| 17-9 _019 | ★ | 56.31 | 1.06 | 20.01 | 5.18 | 0.25 | 0.87 | 1.95 | 8.46 | 5.32 | 0.28 | 100.00 |
| 17-9 _020 | ★ | 57.64 | 0.93 | 19.94 | 4.83 | 0.25 | 0.84 | 1.67 | 8.73 | 4.78 | 0.22 | 100.00 |
| 17-9 _032 | ★ | 56.86 | 1.04 | 20.09 | 5.18 | 0.31 | 0.86 | 1.76 | 8.24 | 5.25 | 0.23 | 100.00 |
| 18-1_014 | ○ | 51.70 | 2.94 | 14.15 | 11.16 | 0.16 | 3.09 | 7.72 | 4.38 | 3.72 | 0.94 | 100.00 |
| 18-5_004 | ◆ | 65.13 | 0.12 | 20.50 | 0.22 | n.d. | 0.00 | 1.82 | 8.32 | 3.97 | n.d. | 100.00 |
| 18-5_013 | ◆ | 72.81 | 0.39 | 15.29 | 1.76 | 0.17 | 0.38 | 1.10 | 4.08 | 4.12 | n.d. | 100.00 |
| 18-5_020* | ◆ | 53.09 | 1.03 | 18.21 | 13.84 | 2.37 | 0.54 | 4.28 | 5.06 | 0.55 | 0.86 | 100.00 |

"a" indicates values averaged over multiple analyses on a single glass shard; * indicates analytical totals below 60%.

The 16.87-16.915 m sample falls between the 1985 and 1986 annual horizons (17.282 and 16.679 m respectively). This sample depth coincides with the end of the austral winter sodium peak and the start of the the sulfate-on-chloride peak, indicating that this sample likely represents snow from the middle of the year (late austral winter to early spring) 1985.

For clarity, we will hereafter refer to these two samples by their year of origin, according to the MBS2023 chronology; 1991 for the 13.28-13.34 m sample, and 1985 for the 16.87-16.915 m sample.

## 4 Volcanic source identification

The majority of sample depth ranges from MBS-Alpha contain sparse and/or heterogeneous tephra shard compositions. However, when plotted on the total alkalis vs. silica diagram (TAS; Le Bas et al. (1986)), the 1991 and 1985 samples contain abundant glass shards (13 and 10 shards respectively), each with largely homogeneous compositions (Fig. 4). We therefore consider these two samples as discrete cryptotephra horizons in the MBS-Alpha core. The remaining samples each contain three or fewer glass shards. Some of these samples do cluster together across sample depths (e.g. cluster of rhyolitic deposits, Fig. 4), however as these shards comprise a wide range of ice core depths, they cannot be correlated as the product of a single eruption event. As none of the remaining samples (aside from the 13.28-13.34 m and 16.87-16.915 m horizons) include more than three shards, for this study, we primarily focus our source attribution efforts on groups with more than three shards of similar composition to ensure reliable correlations (similar to the approach taken by Cook et al. (2022)).



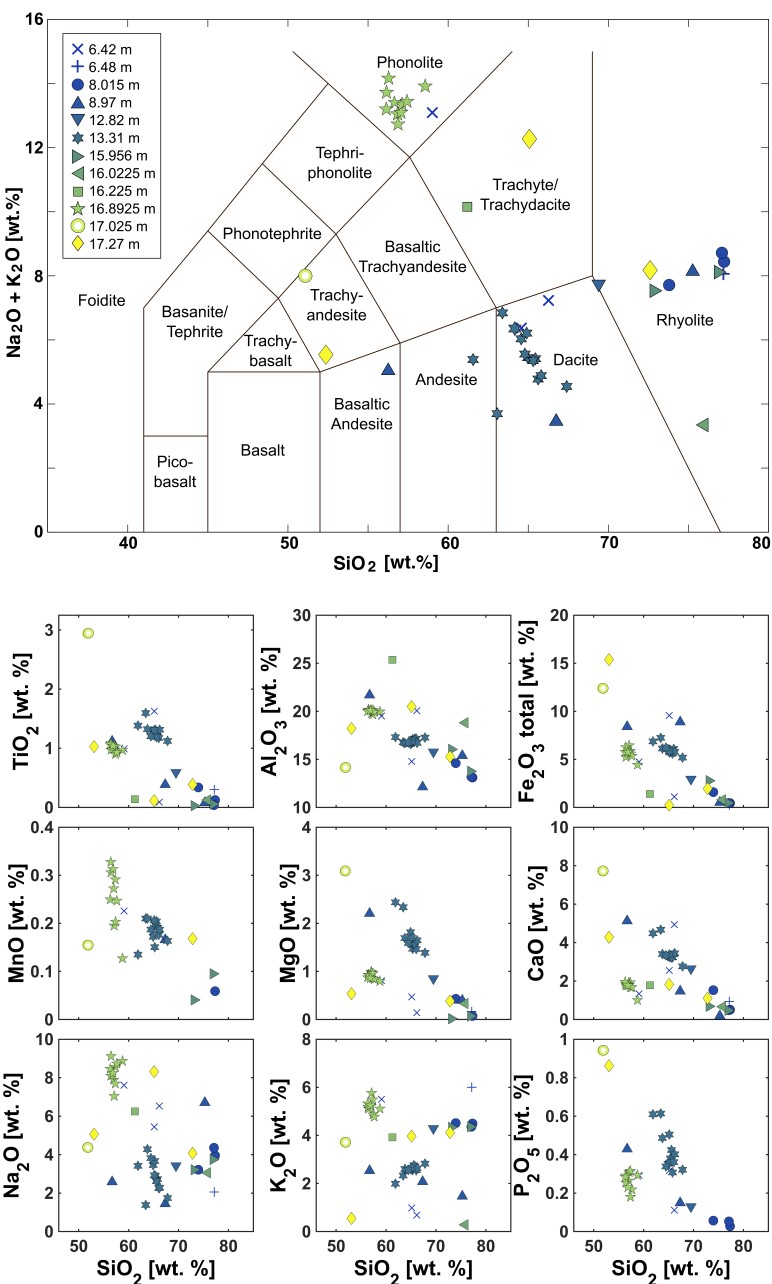

**Figure 4.** Total alkalis vs silica (TAS, Le Bas et al. (1986)) diagram and variation diagrams for all individual glass shards identified in the MBS-Alpha core. Data are normalized to 100% (anhydrous) using the Cl-adjusted analytical total, after Iverson et al. (2017).

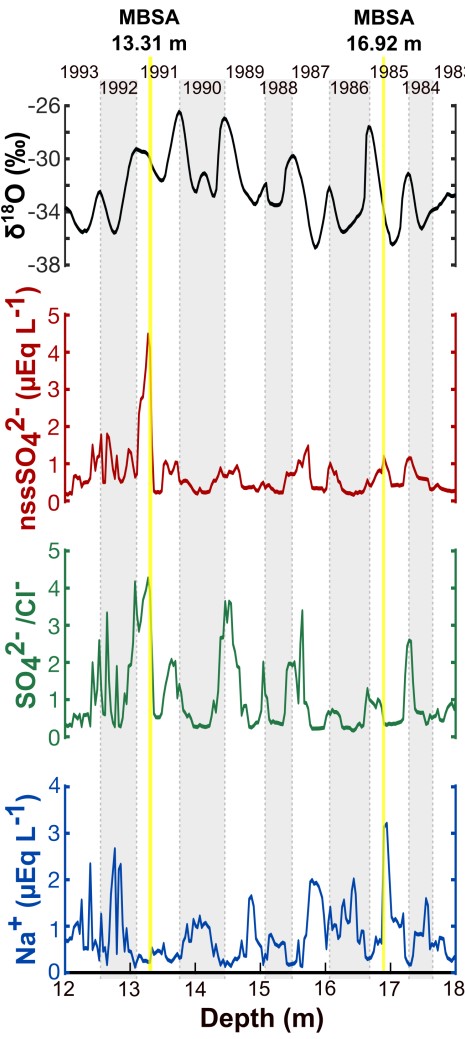

**Figure 5.** Selected seasonally varying glaciochemical species (non-seasalt sulfate (nssSO$_4^{2-}$), ratio of sulfate to chloride (SO$_4^{2-}$/Cl$^-$), and sea salts (Na$^+$)) and isotope ($\delta^{18}$O) measured from the MBS-Alpha core with tephra horizon depths indicated (yellow bars). Annual horizons (MBS2023 January 1 year boundaries, after Vance et al. (2024a)) indicated by vertical dashed lines.

## 4.1 Rhyolite group

A handful of shards originating from a range of sample depths (from 6.48 to 17.27 m depth), form a compositionally similar group of rhyolites. Despite originating from a broad range of depths (and therefore ages), if treated as a group, the composition of these shards cluster together across most major element oxides (Fig. 4). The composition of this cluster is similar to compositions determined for volcanic glass shards produced in the 1991 eruption of Pinatubo (Luhr and Melson, 1996; Tamura and





Nakagawa, 2023). However none of the depths where glass shards of this composition were found correspond within reasonable dating error to the 1991 eruption of Pinatubo, where a sulfate peak is often used as a dating horizon for 1991/1992 in other ice cores.

The rhyolitic glass shard compositions fall within the range of compositions of sparse mid- to late-Holocene rhyolite shards
reported in the B53 and B54 ice cores, which Abbott et al. (2024) correlate with shards identified in the Talos Dome ice core. Narcisi et al. (2012) attribute these Talos Dome rhyolite shards to various extra-Antarctic sources (including Andean and New Zealand sources). Del Carlo et al. (2018) have correlated similar high-K rhyolites with either "extremely evolved" products of West Antarctic intraplate volcanoes (including McMurdo group and Marie Byrd Land volcanoes, not active during the satellite era), but also show similarities to Antarctic samples correlated with South American (Aguilera) as well as South Shetland
Islands volcanoes. Due to the multiplicity of sources attributed to tephra with similar compositions, as well as their sparseness in the MBS-Alpha core, we cannot confidently attribute a single source to this compositional group of rhyolites.

An alternative interpretation of such sparse and heterogeneous samples is to suppose they represent a fraction of the background "dust" signal often seen in ice core analyses (Plunkett et al., 2020; Hutchison et al., 2024; Vallelonga and Svensson, 2014; Delmonte et al., 2013). This background signal could include re-mobilized volcanic ash from a variety of sources, trans-
ported from nearby ice-free areas or other extra-Antarctic sources (Delmonte et al., 2013). Typically volcanic glass shards transported by wind display evidence of reworking e.g. abraded margins (Dunbar et al., 2003). Although textural alteration depends on the duration and conditions of weathering glass shards are exposed to, all glass shards in this group have angular shard-like morphologies that we interpret to be primary textures.

## 4.2  1991 Horizon

The presence of a substantial spike in non-sea salt sulfate (likely volcanic) co-occurring with the 1991 sample bearing multiple glass shards suggests that the tephra found could be associated with the volcanic event that produced the sulfate signal in the ice core. Based on our understanding of the age of the sample , we consider the two primary candidates for the source of the 1991 tephra horizon to be the eruptions of Pinatubo (Philippines, VEI 6) and Cerro Hudson (Chile, VEI 5), both occuring in mid-1991. However we also investigate all other significant (VEI 3 or greater) known tropical and Southern Hemisphere eruptions
from 1991 (Nyamulagira, Democratic Republic of Congo and Lokon-Empung, Indonesia). Additionally, we include other Antarctic and sub-Antarctic volcanic sources (Gaussberg, South Sandwich Islands, and South Shetland Islands), as although there are no known eruptions of these volcanoes during 1991, they are known or speculated possible sources of tephra (primary or reworked) identified in other Antarctic ice cores (Geyer et al., 2023; Narcisi and Petit, 2021).

It is clear from the TAS diagram (Fig. 6) that products of Nyamulagira and Lokon-Empung do not correlate with glass from
the MBS-Alpha 1991 horizon, and neither do the products of Gaussberg or the South Shetland Islands. We therefore rule these out as potential sources. While the South Sandwich Islands do appear to correlate somewhat better on the TAS diagram, when compared with literature values, the 1991 tephra is characterized by significantly higher $K_2O$ values (1.99-3.09 wt.%) than South Sandwich Islands (<1.5 wt.% $K_2O$) for similar $SiO_2$ ranges (Abbott et al., 2024; Pearce et al., 1995). This combined





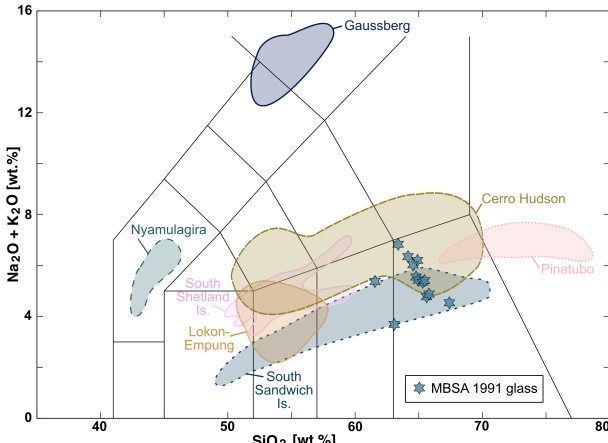

**Figure 6.** Total alkali vs silica diagram (Le Bas et al., 1986) showing the MBS-Alpha 1991 glass together with representative data for possible eruption sources: Cerro Hudson (Haberle and Lumley, 1998), Nyamulagira (Aoki et al., 1985), Lokon-Empung (Dmitrieva et al., 2023), Gaussberg (Salvioli-Mariani et al., 2004), Pinatubo (Luhr and Melson, 1996; Tamura and Nakagawa, 2023), South Sandwich and South Shetland Islands (Narcisi et al. (2005), and references therein).

with no recorded eruptions of the South Sandwich Island volcanoes during this time period, leads us to rule out South Sandwich
Islands as a potential source.

We are left with Cerro Hudson, and Pinatubo, both major 1991 eruptions (VEI 5 and 6, respectively) as potential sources for the glass in the MBS-Alpha 1991 horizon. The products of the 1991 eruption of Pinatubo are primarily rhyolitic (Fig. 7), and have much lower $TiO_2$ (<0.52 wt.%) and FeO (<3.01 wt.%; (Luhr and Melson, 1996; Tamura and Nakagawa, 2023)) than both the MBS-Alpha 1991 glass and the majority of the Cerro Hudson products. Based on the difference in $TiO_2$ and FeO, we
rule out Pinatubo as a potential source candidate for the dacite cluster in the 1991 tephra. The geochemical compositions of the glass shards in the 1991 sample show a strong similarity to the Holocene volcanic products of the Cerro Hudson volcano (Fig. 7).

The major elements of the 1991 dacite tephra match well with Holocene volcanic products of Cerro Hudson. This is seen particularly with the later products from the 1991 eruption, as the composition shifted from basaltic-andesite towards large
volumes of trachyandesite to rhyo-dacite products (Fig. 7,Wilson et al. (2011); Naranjo S. et al. (1993); Kratzmann et al. (2009)). The dacite glass from the 13.3 tephra horizon show a slightly less alkaline composition than some Cerro Hudson products, including glass shards from the 1991 eruption recovered from visible tephra deposits in lake and terrestrial cores from southern Chile (Streeter et al. (2024), Fig. 7). This lower alkalinity is driven largely by a lower concentration of $Na_2O$, while the remaining major element oxides show similar concentrations to the literature values for Cerro Hudson eruption
products (Fig. 7; Fernandez and Bitschene (1993); Haberle and Lumley (1998); Gutiérrez et al. (2005); Kratzmann et al. (2009); Del Carlo et al. (2018); Panaretos et al. (2021); Abbott et al. (2024); Streeter et al. (2024)). While this lower $Na_2O$ value could be due to minimal migration of sodium during EPMA analysis (Gedeon et al., 2000; Humphreys et al., 2006; Kuehn



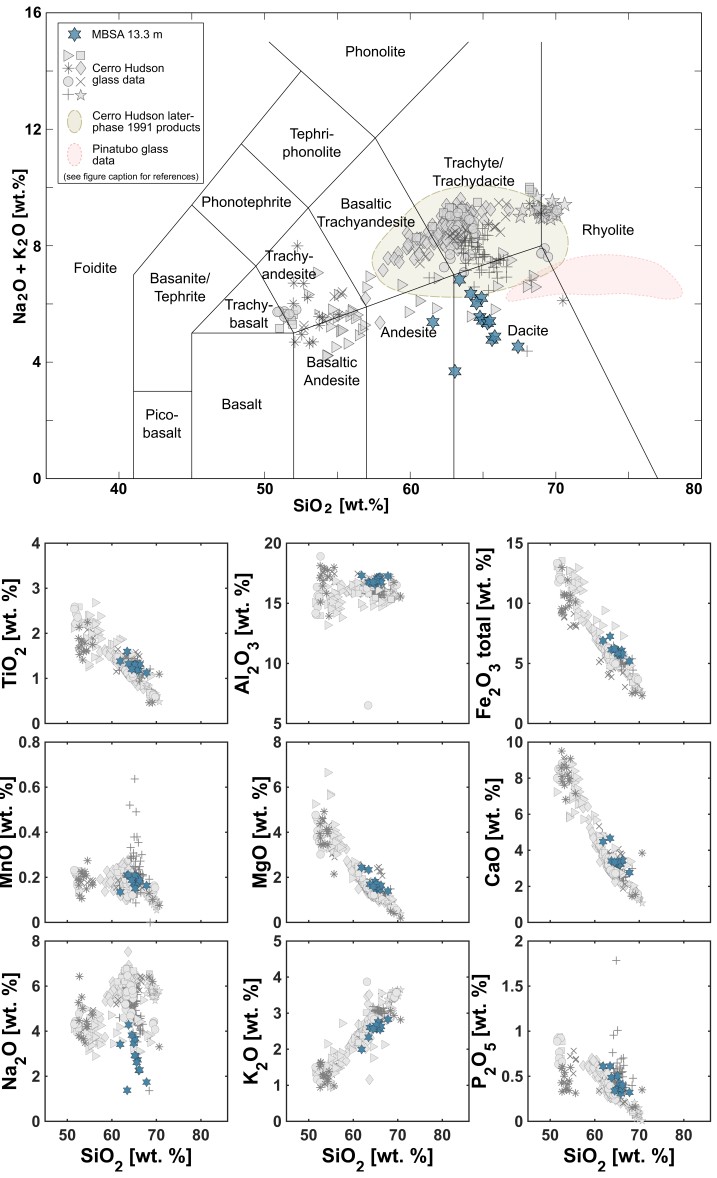

**Figure 7.** Total alkali vs silica diagram (Le Bas et al. (1986)) and individual major element variation diagrams showing the 13.3 tephra horizon from MBS-Alpha (teal hexagram), together with literature values from various Holocene eruptions of Cerro Hudson (grey circles (Fernandez and Bitschene, 1993), triangles (Haberle and Lumley, 1998), crosses (Gutiérrez et al., 2005), squares (Kratzmann et al., 2010), plusses (Del Carlo et al., 2018), stars (Panaretos et al., 2021), asterisks (Abbott et al., 2024), and diamonds (Streeter et al., 2024)). Shaded area indicates characteristic compositions of volcanic glasses from the 1991 eruption of Mt. Pinatubo (Luhr and Melson, 1996; Tamura and Nakagawa, 2023).





et al., 2011), we consider this unlikely, as analytical conditions for EPMA were selected to minimize potential migration of alkali ions for small glass shards, and repeated analysis of secondary standards did not show a loss of sodium over time (see supplementary materials). Based the geochemistry and chronology of this cryptotephra horizon, we propose that the MBS-Alpha 1991 cryptotephra originate from the 1991 eruption of Cerro Hudson.

### 4.2.1 1991 Horizon: Sulfate and tephra deposition timing

The sample depth of the 1991 tephra horizon coincides with the start of a significant increase in the non-sea salt sulfate (nss-$SO_4^{2-}$) chemistry of the MBS-Alpha core (Fig. 5). It is common in ice core tephra studies to see a tephra layer deposited before the corresponding peak in nss-$SO_4^{2-}$ (Burke et al., 2019). The delay between deposition of volcanic ash and volcanic-produced sulfate is variable, based on a combination of distance from the source volcano, scale and plume height of eruptive event, atmospheric transport conditions (Abbott et al., 2024; Plunkett et al., 2023; Koffman et al., 2017; Burke et al., 2019), as well as potential diffusion within the ice (Rhodes et al., 2024). A peak in sulfate that occurs without substantial lag following tephra deposition may indicate more rapid tropospheric transport and shorter residence time of sulfate aerosols, however the relationship is complicated in cases of long range transport from tropical eruptions (Plunkett et al., 2023). For the 520 BCE $HW_6$ eruption of Cerro Hudson, Abbott et al. (2024) identify a time lag of less than one year between elevated microparticle levels (indicative of volcanic ash/tephra deposition) and the peak in sulfate deposition in the East Antarctic Plateau B53 ice core.

In the case of the 1991 horizon, however, the sulfate spike is assumed to be a combined signal from both the Pinatubo eruption (June 12-15, 1991) and the Cerro Hudson eruption (August 8-12, 1991). The 1991 eruption of Pinatubo ejected an approximated 18 - 20 megatons of $SO_2$ into the atmosphere (Guo et al., 2004; Pitts and Thomason, 1993), the Cerro Hudson eruption two months later only produced an estimated 1.5 - 4 megatons of $SO_2$ (Doiron et al., 1991; Constantine et al., 2000; Case et al., 2024). The $SO_2$ produced in volcanic eruptions is oxidized through atmospheric processes, becoming $SO_4^{2-}$ before deposition onto the snow surface, where it is measurable in the ice core (Burke et al., 2019). The order-of-magnitude greater $SO_2$ injection from Pinatubo prevents us from being able to disentangle the Hudson $SO_2$ signal from bulk ice core measurements without analysis of sulfur isotopes (Burke et al., 2019). Despite the difference in $SO_2$ injection, the Cerro Hudson eruption is estimated to have produced a slightly larger volume of tephra-fall deposits than Pinatubo: 2.7 km$^3$ and 1.8 - 2.2 km$^3$ dense-rock equivalent, respectively (Guo et al., 2004; Paladio-Melosantos et al., 1996). Additionally, it is not uncommon for multiple peaks to be seen in ice core sulfate measurements in years following a major eruption, which can be due to seasonal deposition of sulfate or confounding eruptions (Legrand and Wagenbach, 1999; Guo et al., 2004). Due to the confounding sulfate signals, however, without sulfur isotopic measurements, interpretation of the time-lag between Cerro Hudson tephra deposition and bulk sulfate spike is not possible.

### 4.2.2 1991 Horizon: Atmospheric circulation

The ash dispersal and sulfate aerosol cloud produced by the 1991 eruption of Cerro Hudson has been well studied (Doiron et al., 1991; Schoeberl et al., 1993; Constantine et al., 2000; Kratzmann et al., 2010; Case et al., 2024). Use of infrared advanced very





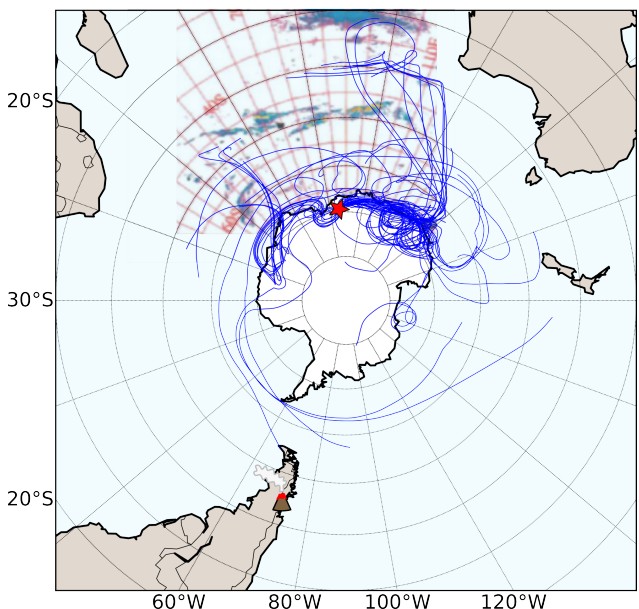

**Figure 8.** Map showing ten-day back trajectories originating every six hours (44 total trajectories) from the MBS site (indicated by red star), from 12 to 22 August, 1991. An overlay of the AVHRR data in blue and yellow shading (image adapted from Constantine et al. (2000)) shows the location of the Cerro Hudson ash cloud on 18 August, 1991. Cerro Hudson indicated by volcano symbol.

high resolution radiometer (AVHRR) and total ozone mapping spectrometer (TOMS) satellite data can be used to investigate volcanic ash and sulfate aerosol distribution, respectively (Doiron et al., 1991; Constantine et al., 2000; Carn et al., 2003). The majority of the ash cloud produced during the Cerro Hudson eruptions was injected near the tropopause, and 90% of the ash cloud was observed to have settled out within a few days (Constantine et al., 2000). AVHRR data show that some part

of the ash cloud was detected as far as Australia within five to six days of the August 15 eruptive phase (Constantine et al., 2000). AVHRR images from 18 August, 1991 show part of the Cerro Hudson volcanic ash cloud positioned over the Kerguelen Plateau, north of the MBS site, lying along the 50°S parallel. On the same day, a smaller fragment of the ash cloud appears to be situated north-west of the MBS site, at approximately 60°S, 45°E (Constantine et al. (2000), Fig 8).

     It has been established that MBS is climatologically linked to the Southern Indian Ocean (Vance et al., 2016), and at-

mospheric back trajectory modeling shows that the MBS region regularly receives air masses passing meridionally via the Southern Indian Ocean, and across the Kerguelen Plateau (Jackson et al., 2023). Ten-day HYSPLIT back trajectories computed every 6 hours originating from MBS for 12 - 22 of August, 1991 (dates chosen to coincide with the Cerro Hudson eruption and observed transport of the ash cloud) show multiple trajectories following a path that could reasonably transport Cerro Hudson ash to MBS (Fig 8). Cluster analysis on 6-hourly trajectories, at 1-hour intervals (resulting in 5 clusters) for

the month of August 1991 shows that clusters representing 44% of the trajectories follow a route that would pass through the





August 18 Cerro Hudson ash cloud. These transport setups would readily transport very fine grained ash (like the glass shards identified in the MBS-Alpha 1991 sample) towards the MBS site.

HYSPLIT trajectory evidence for long-range atmospheric transport of volcanic ash from the 1991 eruption of Hudson volcano to the MBS site together with our geochemical evidence for similarities between the MBS-Alpha 1991 cryptotephera

and Cerro Hudson 1991 tephra provide robust support for our interpretation that the cryptotephra in this layer was produced during the 1991 eruption of Cerro Hudson.

### 4.3 1985 Horizon

There are no globally significant volcanoes on the order of magnitude of Cerro Hudson or Pinatubo recorded in 1985 to help narrow the search for the source of the glass shards identified in the 1985 cryptotephra horizon. The phonolitic composition

of the glass shards, however, are uncommon enough to narrow down the candidates for volcanic source matching. To our knowledge, there are only two phonolitic volcanoes active during the last 1000 years (Global Volcanism Program, 2024). These two volcanoes are the McDonald Islands volcano, located near Heard Island on the Kerguelen Plateau, ~2000 km NNW of MBS, and Mount Erebus, Ross Island, Antarctica, ~2500 km SSE of MBS.

There are no known observations of volcanic activity from the McDonald Islands from their discovery in 1874 until 1997.

Since 1997, only two eruptive events (Stephenson et al., 2005) have been described, with ongoing hydrothermal activity (Spain et al., 2020). However, the eruptive history of the islands is poorly known due to their extreme remoteness, and near-constant cloud cover in the region making satellite observation difficult for most of the year (Fox et al., 2021).

Mount Erebus has been continuously active since at least 1972, with periods of increased activity from 1984 through 1985 (Global Volcanism Program and McClelland, 1986). Reports from nearby Scott Base describe frequent activity, including the

detection of strombolian eruptions, audible explosions, and incandescent ejecta seen from McMurdo Sound, and as far away as Butter Point (70 km from the volcano) (Global Volcanism Program and McClelland, 1984).

The phonolite tephra identified in the 1985 horizon show similarities to the composition of volcanic glasses from both Erebus and McDonald Islands (Fig. 9a).

Few analyses of McDonald Islands volcanic material exist in the published literature (Barling et al., 1994; Leach et al.,

2016). Leach et al. (2016) investigated an obsidian rich phonolitic pumice washed up on a beach in the Chatham Islands, Aotearoa, and characterized it as a product of McDonald Islands, likely from the 1997 eruption. Due to the rigorous matching efforts of Leach et al. (2016), and to expand the available match dataset to include recent eruptive products, we include the Chatham Island sample as McDonald for comparison. Analyses from MBS-Alpha 1985 show a less alkaline composition than that of McDonald Islands, largely driven by a lower concentration of $Na_2O$ (Fig 9). Additionally, MBS-Alpha 1985 has higher

concentrations of $TiO_2$, FeO, and MnO, and generally appears distinct from the literature values for McDonald Islands (Barling et al., 1994; Leach et al., 2016).

The variation diagrams for K, Ti, Al, Fe, Mg, and Ca show that the MBS-Alpha 1985 glass correlates more strongly with literature values for Mt. Erebus (Kelly et al., 2008; Iverson et al., 2014; Harpel et al., 2008) than those of the McDonald Islands,





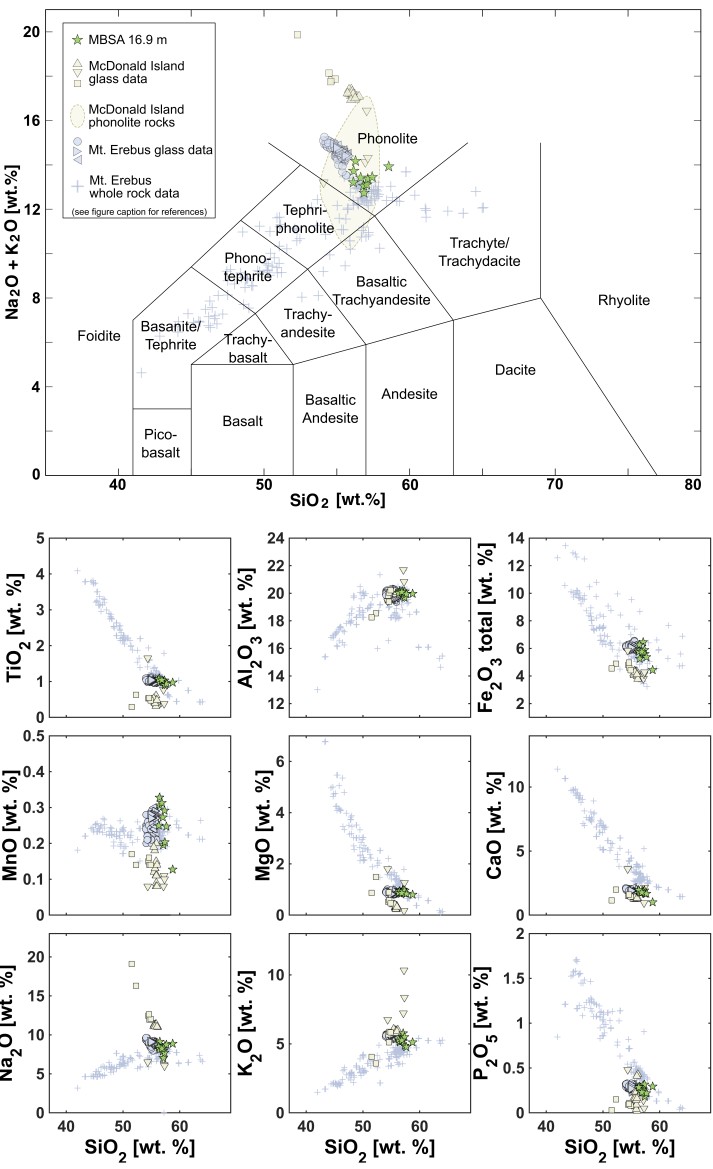

**Figure 9.** Total alkali vs silica diagram (Le Bas et al. (1986)) and individual major element variation diagrams showing the 1985 tephra horizon from MBS-Alpha (green pentagrams, this study), together with literature values for eruptive products from Mt. Erebus (blue right pointing triangles (Kelly et al., 2008), left pointing triangles (Iverson et al., 2014), circles (Harpel et al., 2008), and pluses (Kyle, 1990; Kyle et al., 1992; Martin et al., 2021)) and McDonald Islands (squares (Leach et al. (2016), McDonald Islands samples), upward pointing triangles (Leach et al. (2016) obsidian floater), and downward pointing triangles (Barling et al., 1994)), Shaded area in TAS diagram shows McDonald Islands phonolite rock compositions (reproduced from Clarke et al. (1983) figure 3).





despite the slightly higher $SiO_2$ (Fig. 9). Additionally, when whole rock samples are considered (Kyle, 1990; Kyle et al., 1992;
Martin et al., 2021), the 1985 tephra fall well within the range of $SiO_2$ expected from Erebus.

### 4.3.1  1985 Horizon: Atmospheric circulation

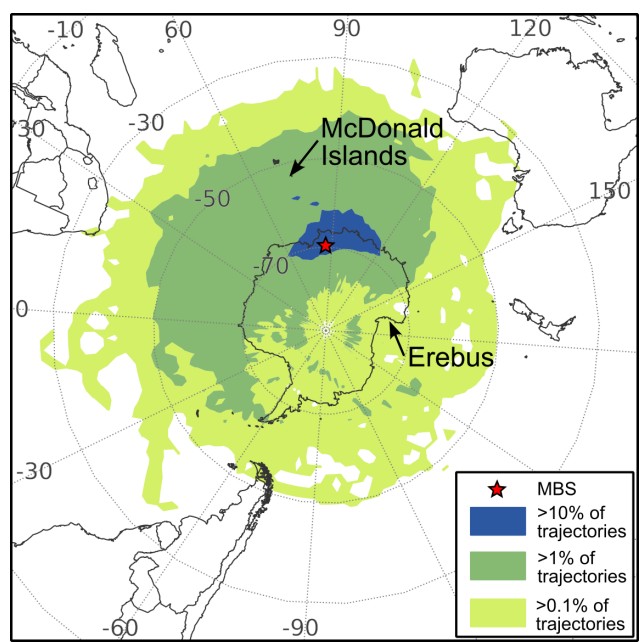

**Figure 10.** Trajectory frequencies for daily 10-day trajectories from May to December 1985. Frequencies calculated as sum of trajectories passing over a 2-degree gridded area, normalized by the total number of trajectories (856 total trajectories).

Using our general age assessment of mid-late 1985 (Fig. 5) as a starting point, we use HYSPLIT to evaluate potential transport to MBS. Analysis of six-hourly ten day back trajectories for May - December 1985 (856 total trajectories) shows that the MBS site received air masses from a wide range of sources during this period (Fig. 10), including the areas around McDonald Islands and Erebus. Back trajectories from MBS passing over McDonald Islands are more frequent (>1% of trajectories), though some trajectories do pass over Erebus (>0.1%).

Potential transport of any volcanic material produced in the Kerguelen Plateau region is supported by our atmospheric trajectory modeling, showing that a significant proportion of trajectories pass over the Kerguelen Plateau en route to MBS. These trajectories likely represent meridional transport of warm air that frequently bring significant amounts of precipitation to the MBS site (Jackson et al., 2023; Vance et al., 2024a). Transport from Erebus likely follows the polar easterlies commonly seen in the region, which align to the mean wind direction at MBS (Vance et al., 2024a).

Our HYSPLIT trajectory analysis indicates that the MBS-Alpha 1985 tephra could have been transported from a variety of sources, including Mt. Erebus and the McDonald Islands. However, the combination of the major element geochemisty presented here and the known volcanic history of the two potential sources lead to our assessment that the 1985 tephra most





likely originated from Mt. Erebus. The potential impact of the identification of Mt. Erebus tephra on our understanding of Antarctic atmospheric transport setups underscores the need for further investigation of the MBS-Alpha 1985 tephra, including trace element analysis.

## 5 Discussion

### 5.1 Ice core dating

Volcanic sulfate and elevated aerosol particle counts have been previously identified in Antarctic ice congruent with the timing of the 1991 eruption of Cerro Hudson (Evangelista et al., 2022; Legrand and Wagenbach, 1999). The MBS 1991 horizon is, to our knowledge, the first geochemically linked identification of Cerro Hudson 1991 glass tephra shards identified in Antarctic ice.

The identification of the 1991 Cerro Hudson eruption in the MBS-Alpha ice core has potential implications for ice core dating of both the MBS ice core array and other East Antarctic ice cores. Volcanic events are commonly used to provide concrete tie-points in ice core dating efforts, however many of these tie-points are based on nssSO$_4$ measurements, rather than volcanic ash layers. Volcanic sulfate aerosols can take months to years to be deposited at a distal ice core site, with elevated sulfate levels in ice core chemistry often apparent for multiple years following the eruption event that produced the sulfate (Lin et al., 2022; Svensson et al., 2020; Plunkett et al., 2023; Burke et al., 2019). Additionally, peaks in volcanic sulfate in ice cores have been shown to be altered through diffusion within the ice as well as layer thinning (Rhodes et al., 2024). Elevated sulfate levels caused by the 1991 eruptions of Cerro Hudson and Pinatubo were measured in aerosol measurements at coastal Antarctic stations and in snowfall across the high Antarctic Plateau over multiple years, until at least the end of 1993 (Legrand and Wagenbach, 1999).

This delay in sulfate deposition and multiple sulfate peak phenomenon, as well as within-core migration of sulfate, can cause discrepancies in ice core dating (Plummer et al., 2012; Sigl et al., 2014, 2015; Crockart et al., 2021). The inability to distinguish the source of nssSO$_4$ deposition between multiple potential source volcanoes without sulfur isotope analysis adds an additional layer of complexity to volcanic matching in ice core dating (Lin et al., 2022; Burke et al., 2019). In the dating of snowpit data from the Dome Fuji site in East Antarctica (Dronning Maud Land), Oyabu et al. (2023) are unable to use the Cerro Hudson/Pinatubo volcanic horizon as a dating tie-point due to the presence of three nssSO$_4$ peaks within a ~0.5 m interval, likely due to seasonal variability in volcanic sulfate deposition, but possibly also from multiple volcanic eruptions.

Due to rapid tropospheric transport, however, volcanic ash can be advected to an ice core site on the order of days to weeks. For this reason, tephrochronology can be a more reliable means of locating volcanic tie-points in ice core dating (Geyer et al., 2023; Cook et al., 2022; Lin et al., 2022). Geochemical confirmation of Cerro Hudson as the source of the 1991 horizon confirms the accuracy of the dating of the MBS-Alpha ice core, as the 1991 sample is appropriately linked to mid-1991 in the current MBS chronology (MBS2023, Vance et al. (2024a)). As the time period around 1991 has previously provided challenges with ice core dating (Plummer et al., 2012; Crockart et al., 2021), the ability to assign a specific date to the 1991





cryptotephra horizon identified here could prove instrumental in verifying future ice core dating efforts, and enable the more accurate characterization of climate proxy-signal links in ice core research.

Several globally significant volcanic eruptions have been identified in the sulfate and/or conductivity record of the 1137-year record of MBS (Vance et al., 2024a; Harlan et al., 2024a). These eruptions have informed the chronology and synchronization of MBS with other Antarctic ice cores (WAIS, Law Dome, and Roosevelt Island, Vance et al. (2024a)) The identification here of extra-Antarctic (Cerro Hudson) cryptotephra in MBS confirms the potential to locate and identify (crypto)tephra from other globally significant eruptions in the deeper ice of MBS. Identification of any of the volcanic events proposed by Vance et al. (2024a) as volcanic tie-points would have critical implications for refinement of the chronology of MBS and other Antarctic ice cores spanning similar time periods.

## 5.2  MBS as tephra archive

Here we have identified and described two cryptotephra horizons in the MBS-Alpha ice core, with 12 dacitic shards identified in 1991, and 10 phonolitic shards in 1985. These tephra horizons highlight the importance of the coastal East Antarctic site as a potential millennial-length archive for future tephrochronology studies. The atmospheric processes that are necessary to transport volcanic material from Mt. Erebus and Cerro Hudson indicate the variability of potential source regions from which MBS could receive and preserve volcanic ash. When considering daily ten day back trajectories over the satellite era, it is apparent that the site receives air masses from almost the entire circumpolar region.

When studies were made to select the site for the East Antarctic ice core that became MBS, one of the criteria was the prevalence of teleconnections to lower latitudes, especially capturing interconnectedness with large scale modes of climate variability (including the Southern Annular Mode and the Indian Ocean Dipole; Vance et al. (2016); Crockart et al. (2021); Udy et al. (2021)), however with thorough back trajectory analysis, it is clear that the climatological transport pathways linking to MBS go beyond its connection with the southern Indian Ocean, and connect the MBS site with a broad range of high latitude airmasses. Together with the evidence of very long-range transport of volcanic material (on the order of thousands to tens of thousands of kilometers), this predicts a great opportunity for further comprehensive tephra sampling from the full length of the ∼300 m (1137 years) of MBS cores to develop a coastal East Antarctic tephrochronology framework covering the last millennium. Such a tephrochronology has the potential to capture products of large scale volcanic events throughout this period, including eruptions such as Krakatau and Tarawera (1884), Tambora (1815), Kuwae (1454), and the 1259 unknown eruption.

## 5.3  Tephra transport pathways

Recent investigations into weather systems like atmospheric rivers and extreme precipitation across the Antarctic continent (Wille et al., 2019, 2021; Inda-Díaz et al., 2021; Baiman et al., 2023; Maclennan et al., 2022) show that these systems bring significant heat and precipitation from the sub-tropics and mid-latitudes to polar sites like MBS over short timescales. The coastal Antarctic site that makes MBS so well suited to be a high resolution record of climate variability (Vance et al., 2016; Crockart et al., 2021; Jackson et al., 2023) also increases its chance of preserving tephra from lower latitudes if eruptions





coincide with these kinds of atmospheric events. Ice core sites like MBS might preserve tephra from older eruptions which, as
with Cerro Hudson, have been previously thought to be unlikely candidates for the East Antarctic site. Our results demonstrate
tephra transport to MBS on the scale of thousands of kilometers by both westerly (from Cerro Hudson) and easterly (from Mt.
Erebus) winds.

The presence of Erebus tephra at MBS also provides important insights into Antarctic atmospheric transport. Transport

from Erebus contradicts to the general understanding that the majority of atmospheric transport to Antarctica follows the
circumpolar westerly winds (Neff and Bertler, 2015). As evidenced by the Hysplit trajectory modeling (Fig. 10), transport
from Erebus to MBS occurs relatively infrequently, and primarily in line with the Antarctic polar easterly winds, which are
linked to the southern edge of synoptic scale mid-latitude low pressure systems. These polar easterlies can be seen in the
mean zonal winds, in line with the snow features seen and conditions experienced during drilling at the MBS site (Vance

et al., 2024a). The identification of MBS-Alpha 1985 tephra as originating from Erebus poses these polar easterly winds as a
potentially overlooked mechanism for particle transport to coastal East Antarctic sites.

Despite our assessment that the 1985 sample originates from Erebus, the prevalence of HYSPLIT trajectories from Mc-
Donald Islands, and our understanding of the transport and precipitation sources via the Southern Indian Ocean leave open
a possibility for meridional transport along this pathway to bring volcanic material from the McDonald Islands and nearby

Heard Island to MBS. Transport from McDonald Islands would be supported by the same processes that transported the 1991
tephra from Cerro Hudson; warm airmasses regularly transported along this meridional pathway, bringing substantial amounts
of snow to the MBS site (Jackson et al., 2023). Investigations into bubble free layer features in ice cores like MBS and the
Law Dome ice core have been studied based on their potential to record temperature changes during and after large events
transporting vapor from low latitudes (Zhang et al., 2023). Further development of this method could be used to help under-

stand specific moisture transport conditions in cases where bubble free layers fall within tephra sample depths, including the
16.87-16.915 m depth (1985) sample, in which one such bubble free layer feature has been identified (at 16.88 m depth).

The identification of Cerro Hudson tephra in MBS ice verifies the proposed meridional link from MBS to the southern Indian
Ocean. While there is satellite image evidence of the Cerrro Hudson ash cloud sitting over the southern Indian Ocean on August
18, 1991 (Constantine et al., 2000), the presence of tephra from the ash cloud implies that at least a small fraction of the ash

cloud must have been transported meridionally southwards for deposition at MBS to occur. This meridional transport is also
validated by Hysplit modelling for the period spanning the 1991 horizon, and Neff and Bertler (2015) find that atmospheric
transport from Patagonia to the Southern Ocean and Antarctica is highly efficient. These meridional wind patterns are especially
prevalent during high precipitation days at MBS (Jackson et al., 2023; Vance et al., 2024a). These high-precipitation events are
likely to bring any airborne volcanic ash to the site, and drive wet-deposition of tropospheric aerosols, the characteristic mode

of deposition at the MBS site (Crockart et al., 2021).

Previous tephra from Holocene eruptions of Cerro Hudson have been proposed in Antarctic ice cores (Kurbatov et al., 2006;
Narcisi et al., 2010, 2012; Abbott et al., 2024). However, the identification of Cerro Hudson in the Talos Dome ice core by
Narcisi et al. (2012) has been refuted, and potential for transport of tephra from low latitudes to Antarctica has been deemed
unfeasible based on proposed inability of air masses to penetrate the Antarctic polar jet stream (Del Carlo et al., 2018). Our





identification of tephra from the 1991 Cerro Hudson eruption in MBS stands in opposition of this claim, and supports studies on the prevalence of rapid, meridional moisture transport and precipitation to coastal East Antarctica (Wille et al., 2019, 2021; Jackson et al., 2023; Turner et al., 2019). Our results confirming meridional transport support the proposed correlations of other East Antarctic ice cores with South American volcanic sources (e.g. Kurbatov et al. (2006); Narcisi et al. (2010, 2012)).

## 6 Conclusions

A thorough sampling of the satellite-era Mount Brown South Alpha ice core yielded >40 volcanic glass shards from 12 samples. Of these, two samples, each with homogeneous compositional clusters of 10 and 14 glass shards, respectively, are proposed to originate from eruptions of Erebus (active throughout the early to mid-1980s) and the 1991 eruption of Cerro Hudson. The identification of two tephra layers in a core spanning only ∼20 m (40 years) is notable for future studies due to the relative sparseness of tephra previously identified in East Antarctic ice.

Identification of tephra at 1985, and its correlation with Mt. Erebus, suggests that the continuing low-level (VEI 1 or 2), Strombolian eruptions occurring at the site have the ability to produce ash that can be transported thousands of kilometers given favorable atmospheric conditions. Given appropriate transport conditions, volcanic material can be transported along the polar easterly winds prevalent at MBS, despite the more frequent events in which warm moist air is advected southwards bringing extreme precipitation to coastal East Antarctica.

Identifying tephra from the 1991 Cerro Hudson eruption in MBS is especially significant for its relevance to ice core dating and core synchronization. The presence of Cerro Hudson tephra shards at 13.28-13.34 m, correlated to 1991, allows definitive confirmation that the MBS-Alpha core is well dated. As this sulfate spike is typically used as a tie point for the early-mid 1990s in Antarctic ice cores, this tephra can confirm or correct the estimate of the important Pinatubo dating horizon. In cases where multiple sulfate peaks are seen around the eruption year, finding Cerro Hudson ash in the core in mid-1991 removes any doubt

about the timing of sulfate deposition at this site. Additionally, this tephra provides evidence of extra-Antarctic volcanic ash passing the Antarctic circumpolar wind belt to be deposited in Antarctica, previously thought to be improbable.

Our findings position coastal East Antarctica and the MBS site as an important potential archive for future tephrochronological work, due to it's high resolution and teleconnections to such a diverse array of low latitude source regions. We suggest a future systematic sampling of the MBS main core for the development of a comprehensive tephrochronology of coastal

East Antarctica. Such a millennial-length tephrochronology would provide a record from a climatologically important region, previously underrepresented in the existing East Antarctic ice core array.

*Data availability.* The MBS-Alpha Geochemistry datasets produced in this study are available as part of the downloadable supplementary files accompanying this manuscript. The MBS2023 chronology (Vance et al., 2024b) is available at http://dx.doi.org/doi:10.26179/352b-6298. MBS surface core chemistry and water isotope datasets (Moy et al., 2024; Crockart et al., 2021) are available at http://dx.doi.org/doi:10.26179/372t-

4q89 and http://dx.doi.org/doi:10.4225/15/58eedf6812621.




*Author contributions.* MH led the study, including developing the sampling plan and writing the manuscript. MH, TV, JF, and HAK concieved the study. MH and EC prepared ice core samples for analysis at the University of Copenhagen with the help of AS. MH performed sample analysis at the Central Science Laboratory at the University of Tasmania. MH, JF, EC, and AS contributed to geochemical characterization of samples. TV leads the Mount Brown South Ice Core project. All coauthors contributed to writing the manuscript.

*Competing interests.* The authors declare no competing interests.

*Acknowledgements.* We thank the members of the Mount Brown South project, including, but not limited to, the team of scientists, technicians, and ice core drillers who made the project possible. Thanks also to Iben Koldtoft for assistance in preparing ice core samples in the freezer laboratory at the University of Copenhagen.

We would also like to acknowledge the contributions of the Central Science Laboratory at the University of Tasmania. The geochemical
analysis of the MBS tephra samples in this work would not have been possible without the analytical support of the electron microscopy and x-ray microanalysis group and the expertise of Sandrin Feig. We would also like to thank Sebastien Meffre for sharing expertise on preparation and mounting of fine grained sample material for microanalysis.

This work is supported by the Australian Antarctic Program Partnership funded by the Australian Government as part of the Antarctic Science Collaboration Initiative program (project ID ASCI000002). This project contributes to an Australian Research Council (ARC)
Discovery Project (DP220100606). MH is supported by the Australian Antarctic Program Partnership. HAK is supported by the Carlsberg Foundation (field trip grant) and the DFF Inge Lehmann (1131-00007B), as well as the H2020 Research and Innovation TiPES project (no. 820970). JF is supported by the Japanese Society for the Promotion of Science through a Postdoctoral Fellowship for Research in Japan.

The Scientific Colour Maps (Crameri, 2023) are used in this study to prevent visual distortion of the data and exclusion of readers with colour-vision deficiencies (Crameri et al., 2020).





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
