# Peer review of "Cryptotephra in the East Antarctic Mount Brown South ice core"

_Climate of the Past, 2024_

## Author Comment (AC1)

**Author comments:**
*We thank the reviewer for taking the time review the manuscript and provide thoughtful and constructive feedback. All comments and suggestions have been addressed, please see the below replies in blue italicized text.*

**Reply to anonymous reviewer (RC1):**

This manuscript presents new tephra measurements in the upper ~20 m of two Mount Brown South ice cores from coastal East Antarctica spanning the recent ~40 years. The authors first conduct a broad, lower resolution investigation of tephra particles in the main MBS ice core before doing more high-resolution geochemical analysis on samples on an adjacent firn core to fingerprint specific eruption sources.

*We thank the reviewer for the thoughtful feedback, and comments which will improve our manuscript.*

Overall, I think the methods and results are described well, and the geochemical data, and therefore ties to known volcanoes, are robust. While the authors identified a wide range of sparse tephras in their samples, the interpretation is focused (in my view, rightfully so) on the two clusters of tephra identified in the two samples with the largest number of shards. For these two events, the authors identify appropriate candidate eruptions and then logically rule out unlikely eruptions to conclude that Cerro Hudson and Erebus as the sources for the tephra layers they identify in 1991 and 1985, respectively. One aspect that could be better presented is the analytical uncertainty of the geochemical measurements- often these are included as crosses on the major element diagrams (bottom 9 panels of Figs. 7 and 9, for example). I'd suggest adding these uncertainties if possible, though the analytical error is in a supplementary data table.

*We agree that a visual representation of the error involved with the measurements would be valuable to the reader. We will update the relevant figures to display analytical error.*

While the tephra data and attribution are robust, I did not find the modeling component or some aspects of the discussion to be as convincing and have the following questions/suggestions for those aspects of the manuscript:

First, the HYSPLIT modeling is used to develop air mass back trajectories for the two eruptions to show they are within the probable source region for the MBS site. Given that long-range transport is implicit to most aerosol records in Antarctica, I do not think that using such back trajectories to show that these volcanoes are within the source region is particularly useful since it seems plausible that nearly all volcanoes in the high latitude Southern Hemisphere could result in tephra deposition in Antarctica. To me, it seems the power of concretely attributing tephra to a specific volcanic source is the possibility for modeling emissions scenarios specific to that volcano. Since these eruptions are relatively well documented down to the specific eruption dates, I think running HYSPLIT in forwards mode with ash emissions for each eruption would be very useful to understand if modeled ash dispersion is consistent with the tephra identified at MBS. It may be worth mentioning that HYSPLIT back trajectories are air mass trajectories and therefore do not specifically consider aerosol transport/scavenging (which would impact tephra), but the forwards dispersion model can be specifically run for volcanic ash, though I am not sure how sophisticated the transport scheme is.

*HYSPLIT was used in back trajectories in both cases, as this is a common way of investigating the sources of air masses received at ice core sites. While we agree that particle dispersion modelling ("ash mode") in HYSPLIT is a powerful tool for investigating ash dispersal, we decided against this for a few reasons. In the case of Cerro Hudson, there are existing satellite observations of the ash and sulfate cloud transport, which represents actual transport of the*

*volcanic ash (e.g. Constantine et al., 2000). Additionally, other studies have used Lagrangian modelling to investigate the ash dispersal from Cerro Hudson (e.g. Kratzmann et al., 2010). We therefore made the decision to refer to these studies in lieu of repeating these analyses. We will update the text to more clearly describe our reasoning to not perform HYSPLIT forward trajectories or ash dispersal modeling for the Cerro Hudson eruption.*

*In the case of the Mt. Erebus, there was continuous eruptive activity at Erebus throughout a number of years in the mid-1980s, and with our sample sizes and the dating constraints of the ice core, we are not able to define an exact single eruption event or specific date that can be pinpointed as producing the tephra found at MBS. Therefore, the Erebus eruptive activity cannot be constrained well enough to glean useful information from HYSPLIT forward trajectories for this eruption.*

*We appreciate these comments and will add more descriptive text to the relevant sections to clarify our reasoning for relying primarily on HYSPLIT back trajectories.*

Second, I did not find that most of the discussion was justified given the limited findings of the manuscript specific to these two modern eruptions. The ice core dating section (5.1) focuses on detailing the uncertainties about ice core sulfate records and presents tephra as a more reliable means of developing tie points. I strongly disagree with this. Volcanic synchronization of ice cores using sulfate has led to the development of extremely accurate ice core chronologies (the volcanic chronologies in Sigl et al., 2014 and 2015 being some of the best recent examples) despite some of the complications associated with these records. While tephra can certainly be a powerful tool to identify specific eruptions in certain ice core records, because it is an insoluble particle it will always be much more heterogeneously distributed in the atmosphere than sulfate and therefore have much more limited utility for synchronizing geographically widespread ice core records. Furthermore, low tephra counts, reworked sparse tephras, and large sample volume required for analysis can hinder cryptotephra work. Maybe this section would be more meaningful if the authors identified these specific tephras in all four cores at MBS as well as in other regions of Antarctica to show a viable widespread signal, but identifying these shards in a single shallow core does not justify this broader discussion of ice core dating.

*Thank you for the feedback. We agree that ice core chronologies based on volcanic sulfate records can be incredibly robust and accurate! Our intent is not to call into question ice core chronologies that rely on volcanic sulfate. We aim only to highlight the potential of tephra in ice cores as a strong signal in the case of the MBS array (for which sulfate-based volcanic synchronization was used in chronology development as described in Vance et al., 2024) and similar cores. We will rephrase the text to clarify that these cryptotephra horizons can be a valuable tool in ice core dating in addition to and strengthening the very robust sulfate-based ice core chronologies.*

*Additionally, while the heterogeneity of tephra deposition across the ice sheet is always a factor in tephrochronology work, we do not see this as an explicit limitation of what we aim to present in this manuscript. The finding of tephra from the two specific eruptions outlined in this work (Hudson 1992 and Erebus 1985) provide concrete tie-points for the ice core presented here (MBS-Alpha). While we only aim to provide concrete dating horizons for the MBS-Alpha core, there is a high likelihood that these volcanic horizons would also be seen in the nearby cores (Bravo, Charlie, and Main) due to their proximity with MBS-Alpha. While this work cannot speak directly to the dating of the MBS-Main core (or other ice cores across Antarctica), we do see value in discussing ice core dating more broadly, as this represents the first attempt at tephrochronology work using the MBS cores, and MBS represents a region that is demonstrably underrepresented in the broader Antarctic ice core landscape. We appreciate the feedback*

*provided here, and we will revise the text to clarify the positioning of this work as a proof-of-concept for the strong potential of tephrochronology to provide concrete dating tie-points for the remaining MBS-array cores in the future.*

Lastly, sections 5.2 and 5.3 seemed almost too hypothetical to be meaningful. Both sections describe how a longer MBS cryptotephra could be insightful and potentially linked to large-scale climate and teleconnections, which doesn't particularly relate to the findings of this study.

*We disagree that the atmospheric teleconnections of the MBS site are not relevant to the study, as our understanding of these teleconnections is what informed our decision to use the MBS cores as a tephrochronological archive in the first place. These teleconnections (and associated large-scale meridional atmospheric transport to coastal East Antarctica) play a significant role in what aerosol/particle material gets transported to MBS, and similarly, the likelihood of the MBS cores to be a strong potential tephra archive. We will expand the text of these two sections to better describe the validity and relevance of the MBS site's teleconnections to the lower latitudes.*

Additionally, the authors do not even show that the tephra record from the main MBS record is valid given the limited amount of available sample. All tephra results in this manuscript are from the shallow core where they were able to obtain larger sample volumes, so the sampling approach presented in this study is not applicable to the main MBS core. Overall, I suggest the authors reframe the discussion to be more pertinent and specific to their results presented in the manuscript.

*While the study did use MBS-Alpha due to the large sample volume readily available for this work (part of a PhD project), perhaps we neglected to provide the required nuance to our description of the limitations in availability of MBS-Main material. The limitation in available material at the time of the study was in part due to where the cores were located at the time the study took place. While analytical work was undertaken at the University of Copenhagen as part of a PhD study, there is an additional remaining archive located in Tasmania. The use of the smaller samples available in Copenhagen at the time was chosen both based on sample availability and as a preliminary screening and proof of concept for better understanding the potential of the MBS array cores for tephra studies. While we only present geochemical analysis of the MBS-Alpha core, based on the sampling conducted here, we encourage future tephrochronology work from the remaining archive sections of the MBS-Main core.*

*While the 2 cm$^3$ it is a small sample size, it is not uncommon for ice core tephrochronology studies to rely on samples of a similar size for tephra sampling (e.g. ~2 cm$^3$ in Cook et al., 2018). We agree that the small quantity of sample availability from MBS Main may present potential limitations to future tephra studies from this core, however we do not see this as detracting from the validity of our findings. This investigation of tephra in the MBS core array represents one of the first studies investigating tephrochronology from a millennial-scale ice core drilled in the coastal East Antarctic region. MBS, together with the Law Dome ice cores, provides insight into a region of Antarctica underrepresented by other ice core studies. While, as you say, the minimal sample remaining may limit a future tephrochronology study of MBS-Main (though, as indicated above, 2 cm$^3$ is not an uncommon sample availability for ice core tephra studies), we present this work as an argument for the importance of the region for future tephrochronological studies.*

Minor notes

Page 2 line 53: Very nitpicky, but I'd say a 295 m ice core is intermediate, not deep.
*We will update our terminology accordingly.*

Page 3 line 64-65: I don't think you justify why MBS would be more advantageous than other Antarctic sites for tephra studies. It seems like a coastal site could be even more complicated than interior sites as it would be more influenced by marine biogenic sulfur complicating the link to sulfur peaks and a coastal site presumably would be closer to exposed land and therefore sources of reworked tephra?

*These are good points. While there may be challenges associated with the coastal site due to marine biogenic sulfur, we understand one of the biggest advantages of the MBS site (similar to sites like Law Dome) as a tephra archive is its strong atmospheric links to higher latitudes as compared to more interior Antarctic sites.*

Page 4 line 75: Define IE (I don't think I saw ice equivalent anywhere else before this).

*We will update the text accordingly.*

Page 5 lines 87-89: what do you mean by this? Weren't all depths sampled in the MBS main core and used to guide more detailed analysis on the shallow core? How does that link to moisture sources?

*Not all depths were sampled from MBS-Main. We used potential moisture transport as one of the criteria for our original sample selection from MBS-Main. We will update the manuscript to make this clear in the text.*

Page 6 section 2.3- this seems like a very long-winded way to say that approximate timing of eruptions was guided by linear interpolation between austral summer peaks in sulfate/Cl and austral winter peaks in Na.

*Our intention in section 2.3 is to highlight that although we do approximate ages guided by a linear interpolation between summer/winter peaks, the coastal nature of the MBS cores means that linear interpolation may misrepresent the sub-annual dating due to its inability to capture the seasonal scale accumulation variability at the site. We will revise the section for brevity and clarity to better communicate this.*

Page 7 line 163- any particular reason that backtrajectories initiated at 1500 m AGL?

*Back trajectories were initiated at 1500 m AGL, which at the MBS site, corresponds to approximately 3500 m ASL. A starting height of ~1500 above the ice core site was selected to reduce impacts on the trajectory due to interaction with the surface topography of the surrounding area and minimizes the chance of the long (10-day) trajectories "hitting the ground" and losing accuracy.*

Page 16 Fig 7- Why is the polygon/region on the TAS diagram for Cerro Hudson a different shape on this plot than on Fig 6?

*The polygon in Figure 6 represents the full range of literature values for Cerro Hudson products. The polygon in Figure 7 represents specifically products from the later phase of the eruption, when the composition of the erupted products shifted towards trachyandesite and rhyo-dacite composition. This is briefly described in section 4.2 (lines 269-270) and specified in the figure legend of Fig 7. We will add text to the figure captions to clarify this.*

Page 17 line 306- how would sulfur isotopes allow for disentangling two coincident eruptions that appear as a single mixed sulfate peak in the ice core record? I'm not sure that is possible. I'd just remove the mention of sulfur isotopes or explain it.

*Burke et al. (2019) present a method of distinguishing tropical and extra-tropical eruption signals based on sulfate isotope fractionation due on the stratospheric height of the sulfur injection. While Burke et al. (2019) do not extend this to directly compare the sulfate signals of the Pinatubo and Cerro Hudson eruptions, we see this method as a promising tool that could be applied to these eruptions.*

Page 18 Fig 8- I'm not sure the AVHRR data appeared as intended. It seems extremely blurry.

*This is the original resolution from the figure reproduced from Constantine, et al. (2000). We will do our best to update the figure with better resolution and clarity.*

Page 21 Fig 10- Why the switch to trajectory frequencies instead of backtrajectories? I actually find this approach more useful than individual backtrajectories so I wonder how this would look on Fig 8.

*Trajectory frequencies were selected for Figure 10, as we do not know an exact eruption date for the eruptive activity of Erebus that produced the material found in MBS, and thus ran back trajectories over a greater period of time (856 trajectories for Figure 10, compared with 44 in Figure 8). The visual representation of the 44 individual trajectories is much clearer, and better represents of the actual trajectory paths when plotted as individual trajectories as compared to trajectory frequencies. We are of the opinion that the use of trajectory frequencies in Figure 10 better captures the uncertainties associated with the eruption timing and ice core dating in the case of the Erebus eruption. We will add text in the relevant sections to better describe why the specific trajectory plots were presented as such.*

Page 25 line 481- be consistent with the number of shards. These are different than Table 1 and those stated on lines 417-418.

*This is a mistake and will be corrected accordingly.*

***References:***

*Burke, A., Moore, K. A., Sigl, M., Nita, D. C., McConnell, J. R., and Adkins, J. F.: Stratospheric eruptions from tropical and extratropical volcanoes constrained using high-resolution sulfur isotopes in ice cores, Earth and Planetary Science Letters, 521, 113–119, https://doi.org/https://doi.org/10.1016/j.epsl.2019.06.006, 2019.*

*Constantine, E. K., Bluth, G. J. S., and Rose, W. I.: Toms and Avhrr Observations of Drifting Volcanic Clouds from the August 1991 Eruptions of Cerro Hudson, pp. 45–64, American Geophysical Union (AGU), https://doi.org/https://doi.org/10.1029/GM116p0045, 2000*

*Cook, E., Portnyagin, M., Ponomareva, V., Bazanova, L., Svensson, A., and Garbe-Schönberg, D.: First identification of cryptotephra from the Kamchatka Peninsula in a Greenland ice core: Implications of a widespread marker deposit that links Greenland to the Pacific northwest, Quaternary Science Reviews, 181, 200–206, 2018.*

*Kratzmann, D. J., Carey, S. N., Fero, J., Scasso, R. A., and Naranjo, J.-A.: Simulations of tephra dispersal from the 1991 explosive eruptions of Hudson volcano, Chile, Journal of Volcanology and Geothermal Research, 190, 337–352, https://doi.org/10.1016/j.jvolgeores.2009.11.021, 2010.*

*Vance, T. R., Abram, N. J., Criscitiello, A. S., Crockart, C. K., DeCampo, A., Favier, V., Gkinis, V., Harlan, M., Jackson, S. L., Kjær, H. A., Long, C. A., Nation, M. K., Plummer, C. T., Segato, D.,*

Spolaor, A., and Vallelonga, P. T.: An annually resolved chronology for the Mount Brown South ice cores, East Antarctica, Climate of the Past, 20, 969–990, https://doi.org/10.5194/cp-20-969-2024, 2024.

---

## Author Comment (AC2)

**Author comments:**
*We thank the reviewer for taking the time review the manuscript and provide thoughtful and constructive feedback. All comments and suggestions have been addressed, please see the below replies in blue italicized text.*

**Reply to anonymous reviewer (RC2):**

The manuscript by Harlan et al. presents a complete and thoughtful investigation of cryptotephra preserved in Mount Brown South ice core layers. They assess the presence of volcanic ash in ice layers, propose potential volcanic sources and aim to characterise transport pathways in a data-sparse region of Antarctica. This manuscript is interesting and relevant and should be considered for publication in Climate of the Past. However, some points need to be discussed before acceptance for publication in Climate of the Past.

> *We thank the reviewer for the positive review and thorough comments, which will strengthen our manuscript.*

Main comments:

- A low-resolution screening was first applied to identify the presence of tephras in bulk samples which analysed a small cross-section (2cm^2) of the MBS-Main core. Then, only samples with tephras were explored in the same layers of MBS-Alpha, this time analysing a larger cross-section (10cm^2). The manuscript discusses in depth the layers with cryptotephras in MBS-Alpha, however, there is no mention of the number of cryptotephras found in MBS-Main or discussion about potential misses of cryptotephra layers in MBS-Main due to the small cross-section analysed. Presenting the results from MBS-Main and discussing potential limitations due to the small cross-section used for the low-resolution screening will help to improve the manuscript.

> *While we agree that presenting the full results from the MBS-Main sampling would strengthen the results here, there were a number of complications with the MBS-main analysis. Due to the small cross section outer edge sample available, we were not able to follow adequate sample cleaning/decontamination procedures. We were therefore not able to constrain the potential for contamination both in terms of potential transfer from one core depth to another and from general lab contamination debris obscuring potential tephra and limiting our ability to form robust identifications. Additionally, due to the sample preparation methods used for the MBS-Main samples, we were not able to reliably section the tephra grains for SEM/EPMA analysis and were thus required to rely only on optical microscopy for these samples. We will update the text to clarify these limitations and provide as much detail from MBS-Main as we are able. Despite these limitations, we were able to use the samples from MBS-Main as a valuable screening tool guiding our subsequent resampling of the MBS-Alpha core.*

> *We will clarify the limitations faced with the MBS-Main samples in the text, and better describe the valuable role they played in guiding our high-resolution sampling strategy.*

- The manuscript describes the method used to sample cryptotephras. Samples were melted inside Whirl-pack bags and then centrifuged inside centrifuge tubes. While moving samples with low-concentration insoluble particles from one container to another, there is always a risk of leaving some of these particles behind, potentially biasing the sample that will finally be analysed. Were there any measures in place to ensure that cryptotephras were not left behind while preparing the samples? Adding this information or including this subject as a potential limitation will help improve the manuscript.

*We took careful precautions to ensure the transfer of as much material as possible by rinsing all whirl-pak bags and glass/plasticware (including centrifuge tubes and transfer pipettes) multiple times with ultrapure water to capture as many particulates as possible from the samples. We will update the methods with more detail on the steps taken to prevent loss of tephra in the sample preparation.*

\- The manuscript discusses the presence of small numbers of cryptotephras found in scattered MBS-Alpha layers. The temporality of these layers indicates these particles are not linked to specific eruptions. Based on this evidence, the authors classify them as background cryptotephra. While these particles could be background cryptotephra, in the absence of more information, they could also be contamination introduced while handling or in the lab environment. Were there any measures or protocols followed while processing the samples which can ensure that those particles were originally inside the ice? Adding this information or including this subject as a potential limitation will help improve the manuscript.

*Careful decontamination processes were taken during the cutting and bagging of the MBS-Alpha ice samples, which were not possible with the MBS-Main samples. MBS-Alpha samples were cut and decontaminated (removal of the outer layer of ice from each sample using a ceramic blade as described in line 115) under a laminar flow hood in the cold room. The cutting surface and blades were wiped clean with isopropanol between each individual ~5 cm sample, and both the ceramic blade used and the cutting surface (a clean, new plastic cutting board) were changed and washed with ultrapure water between samples from each meter of core (e.g. a freshly washed blade and cutting board were used for samples from MBS-Alpha 14, and then washed before decontamination of samples from MBS-Alpha 15).*

*In addition to the decontamination procedures, we also chose to focus our source identification only on samples containing multiple (>3) glass shards of reasonably homogeneous composition, in order to minimize the chances of making source identification claims about artificially introduced particles (contamination).*

*We will include text (either as a supplementary methods document or directly in the text) to clarify the steps taken to prevent contamination between samples.*

\- Throughout the manuscript, it is highlighted that MBS has strong teleconnections with low latitudes and is well connected with the southern Indian Ocean. If MBS is so well connected with the Southern Indian Ocean and low latitudes, why are there no cryptotephra layers found for the mentioned Heard Island eruption (1997) and Pinatubo (1991)? Currently, there is no discussion in the manuscript addressing this issue. The unexpected presence of Erebus cryptotephra is discussed and highlighted as an anecdotic case. However, the wide presence of unexpected Erebus cryptotephra highlights more the absence of the eruptions that "should" have been present in MBS layers. The absence of these expected eruptions highlights the transport of cryptotephra to MBS may be highly biased by the occurrence of favourable atmospheric conditions. Incorporating this topic into the discussion section will help improve the manuscript.

*Thank you for raising this point. We agree, and were also surprised to find such robust tephra from Erebus and none from a more proximal and "obvious" source like Heard Island! We agree that this highlights the significant role of favorable atmospheric transport conditions in what tephra get deposited at a specific site. We will expand the discussion section to address this.*

Minor comments:

Line 26: for reference, specify the distance you are considering when classifying something as distal or ultra-distal.

*We will add definitions to the text to this end.*

Line 37-38: The Chichon (1982) Eruption is mentioned here but not mentioned again throughout the text. Given its magnitude and relevance, I thought this eruption was going to be discussed as a potential source for the 1985 layer.

*While signals from El Chichon 1982 eruption have been reported in Antarctic ice, we do not consider it a potential source as the dating of the MBS cores is robust, with uncertainties less than 3 years at this depth, and we would expect tephra deposition to occur on a much shorter timescale than 3 years.*

Line 44: higher resolution or higher accumulation?

*Higher accumulation. We will correct the text accordingly.*

Line 48 and throughout the text: usually the satellite era is considered as the interval from 1979 to present

*We will update the text accordingly.*

Figure 1: The text mentions many sites which are not included in this map (e.g. Siple, Vostok, Talos, Law Dome, Puyehue). I suggest incorporating them for the reader to understand where those sites are located.

*We will update the map to include additional sites relevant to the text.*

Lines 73-74: specify that the units are "meters depth" for all cores, as done for the MBS-Main core

*We will correct this.*

Lines 75-76: what does IE stand for? Ice equivalent? Please, specify.

*Yes - IE is ice equivalent here. We will update the text accordingly.*

Lines 79-84 and throughout the text: Please, specify that you are referring to "stable water isotopes"

*We will update the text throughout to specify stable water isotopes.*

Line 90: Please specify if you refer to marine fertilization or soil fertilization

*We are referring here to ocean fertilization. We will specify this in the text.*

Line 91: towards

*We will correct this.*

Lune 94: ~20 meters depth

*We will correct this.*

Line 106: please, add further details about the containers where the samples were placed. Were those bottles new? Were they pre-cleaned?

*As described in the previous comment above, we will expand the text in the methods section to better describe the decontamination and sample preparation procedures used.*

Figure 2a: apparently there is no point (4) in the diagram

*Thank you for pointing this out. We will correct the figure accordingly.*

Line 147: Please, specify which 14 major and minor elements were measured

*We will update the text to specify.*

Line 157: the text specifies that totals below 60% are considered with caution. It would be useful if those samples were specifically labelled in the text and figures

*The few (5 total) samples with totals below 60% are indicated in Table 2, and only one of these low-total samples are included in one of the tephra horizons that we identify a source correlation for (17-9_009). We will update the text to include wording that specifically indicates which samples have low analytical totals and refer the reader to the supplementary information for the analytical totals of all provided measurements.*

Line 163: Is there a particular reason why using 1500 AGL instead of other elevation?

*Back trajectories were initiated at 1500 m AGL, which at the MBS site, corresponds to approximately 3500 m ASL. A starting height of ~1500 above the ice core site was selected to reduce impacts on the trajectory due to interaction with the surface topography of the surrounding area and minimizes the chance of the long (10-day) trajectories "hitting the ground" and losing accuracy.*

Line 169: The line starts stating that there were glass shards in 48 out of 70 samples, however, after SEM analyses, this number fall to 29. This suggests that the initial 48 samples had, in fact, "potential glass shards". Please, consider correcting this at the beginning of the paragraph

*We will update the wording to specify potential glass shards here.*

Figure 5: there are no details about how the nssSO4 was calculated. if it was calculated using Na+ or Cl-

*The non-sea salt sulfate was calculated following Plummer et al., (2012): $nssSO_4^{2-} = [total\ SO_4^{2-}] - (0.1201 - r) \cdot [Na^+]$. We will add text to section 3.3 to specify the calculation used.*

Line 261: for temporal reference, consider adding the dates when the eruptions happened

*We will add the respective dates for the two eruptions.*

Line 291: identified, instead of identify

*We will update the text accordingly.*

Lines 309-331: If MBS is so well connected with the Indian Ocean, why are there no cryptotephras from Pinatubo or Heard island?

*We were also surprised to find such robust tephra from Erebus and none from a more proximal and "obvious" source like Heard Island, or a globally significant eruption like Pinatubo. We argue that this highlights the significant role of synoptic-scale favorable atmospheric transport conditions in what tephra get deposited at a specific site. We will expand the discussion section to address this.*

Line 398: were instead of are

*We will update the text accordingly.*

Line 432: Consider specifying a year for Krakatau and a year for Tarawera, as they happened during different years

*Thank you for catching this error! We will update the text to include the correct eruption years for the specific eruptions.*

Line 442: consider adding "is viable" after MBS

*We will update the text accordingly.*

Lines 444-470: Evidence presented in this manuscript supports Erebus cryptotephra made it to MBS. Did other, closer cores (e.g. Talos, Taylor, Gv7) have 1985 Erebus cryptotephra?

*While we are not aware of other cores containing (crypto)tephra from the 1980s eruptive activity of Mt. Erebus, there are several potential correlations of tephra to other periods of Mt. Erebus eruptive activity in more proximal sites (including Narcisi et al., 2010, Iverson et al., 2014, and Harpel et al., 2008)*

Line 463: consider replacing "satellite image evidence" with "remote sensing evidence"

*We will update the text accordingly.*

Line 471: tephra or cryptotephra?

*Cryptotephra - this will be specified in the text.*

Line 481: The line states that there is a cluster of 14 shards, but Table 1 shows only 13

*This is a mistake and will be corrected in the revised text.*

*References:*

*Harpel, C., Kyle, P., and Dunbar, N.: Englacial tephrostratigraphy of Erebus volcano, Antarctica, Journal of Volcanology and Geothermal Research, 177, 549–568,*

*https://doi.org/https://doi.org/10.1016/j.jvolgeores.2008.06.001, volcanology of Erebus volcano, Antarctica, 2008.*

*Iverson, N. A., Kyle, P. R., Dunbar, N. W., McIntosh, W. C., and Pearce, N. J. G.: Eruptive history and magmatic stability of Erebus volcano, Antarctica: Insights from englacial tephra, Geochemistry, Geophysics, Geosystems, 15, 4180–4202, https://doi.org/https://doi.org/10.1002/2014GC005435, 2014.*

*Narcisi, B., Petit, J. R., and Delmonte, B.: Extended East Antarctic ice-core tephrostratigraphy, Quaternary Science Reviews, 29, 21–27, https://doi.org/10.1016/j.quascirev.2009.07.009, 2010.*

*Plummer, C. T., Curran, M. A. J., van Ommen, T. D., Rasmussen, S. O., Moy, A. D., Vance, T. R., Clausen, H. B., Vinther, B. M., and Mayewski, P. A.: An independently dated 2000-yr volcanic record from Law Dome, East Antarctica, including a new perspective on the dating of the 1450s CE eruption of Kuwae, Vanuatu, Climate of the Past, 8, 1929–1940, https://doi.org/10.5194/cp-8-1929-2012, 2012.*

---

## Author Comment (AC3)

**Author comments:**
*We thank the reviewer for taking the time review the manuscript and provide thoughtful and constructive feedback. All comments and suggestions have been addressed, please see the below replies in blue italicized text.*

**Reply to anonymous reviewer (RC3):**

I enjoyed reading about the novel approach applied to the Mount Brown South core array to discover two cryptotephra horizons in the MBS-Alpha surface firn core, and the integration of seasonal glaciochemical signals, atmospheric modelling, and geochemical fingerprinting to support the proposed source attributions. The manuscript presents a thorough method for sample preparation and a well-considered rationale for source attribution that acknowledges the challenges of identifying (crypto)tephra in Antarctic ice cores. The reported discovery of cryptotephra horizons derived from the 1991 CE Cerro Hudson (Chile) and 1985 CE Mt. Erebus (Antarctica) eruptions establishes promising new satellite era time-stratigraphic markers for East Antarctica, and highlights the value of future investigation of the MBS-Main ice core to further enhance the emerging tephrochronological framework for Antarctica.

> *We thank the reviewer for acknowledging the value of our work, and for providing comments which will improve the manuscript.*

Overall the content of this manuscript is logically presented and well-written, however there are some minor errors and inconsistencies that should be addressed to improve the presentation of key data and subsequent discussion. In particular, I would like to draw attention to the following areas:

Core selection and justification: It is unclear from the main text (sections 2.1-2.2) why the MBS-Alpha firn core was selected for examination instead of MBS-Bravo or MBS-Charlie. Was this selection based on differences in accumulation rates? The Figure 1 caption suggests that the MBS-A core was chosen "due to larger available sample volume", please clarify.

> *Thank you for raising this concern. As the core selection justification was also raised by other reviewers, it is clear that this can be clarified further in the text of the manuscript. We will therefore revise the methods (Section 2) to include a clearer justification of the sampling procedure chosen and the reasonings behind our use of the MBS-Main and -Alpha cores. We will be sure to include the following information:*
> - *Amount of MBS-Main ice remaining and available for sampling and how this impacted the samples collected from MBS-Main*
> - *Challenges faced with the sample mounting procedure used for the MBS-Main core*
> - *Sample availability of the MBS-Alpha core*
> - *Comparison between MBS-Main and MBS-Alpha (based on pre-existing chemistry and isotope data) and justification for using MBS-Alpha to infer about the tephra archive potential of MBS-Main.*

Sample selection and SEM-EDS data: Please clarify why 19 samples from the MBS-Alpha firn core were not selected for further analysis by EPMA-WDS (Figure 2a-7; lines 169-171). Was there glass identified in these samples but deemed too small for analysis? Are the results of the SEM-EDS analyses conducted (Figure 2a-7) available, and if so do they concur with the EPMA-WDS results presented for samples that progressed (Figure 2a-8)? Considering the use of a small beam diameter of 2 μm for EPMA-WDS please clarify why a further 17 samples were not analysed by EPMA-WDS (Figure 2a-8), were there no glass shards identified or was the material too small for analysis (lines 171-172; cf. line 179)? Define what is meant by "tephra grains of suitable size" (line 123), which would help justify the reasons for why these samples may have been suitable for further analysis.

*The questions raised here make it apparent that more clarity is needed throughout section 2.2 on the analyses conducted, how the samples were chosen for each analysis, and the resulting data from each step. We will add text in the relevant sections that more thoroughly describes the sample selection steps and procedure, including the following:*
- *Detailed description of what analyses were conducted on each sample*
- *Clearer definition of the criteria used at each step of sampling (e.g. glass shard size criteria for EPMA-WDS)*
- *Which samples were not selected for further analysis and why.*

EPMA-WDS analytical conditions and Na loss: Please provide some further clarification and justification (line 145) as to why these analytical conditions were used for EPMA-WDS. For example, Innes et al. (2024) propose that a 3-μm EPMA beam is suitable for use on all glass compositions provided that the beam current is reduced to 1 nA. Given the use of the "broad beam overlap" method of Iverson et al. (2017) why not use a larger beam of $\geq 5$ μm in diameter for larger microparticles identified? This would have reduced the need for multiple analyses of individual glass shards from MBS-A sample 14-1 that likely increased the likelihood of Na loss.

*While the best practices proposed by Innes et al. (2024) are robust and a good framework for conducting EPMA analysis on fine grained tephra, our analyses were performed before these recommendations were published. EPMA analytical conditions were selected based on the recommendation of the expert microprobe staff technicians we worked with, with the aim of obtaining as robust or analytical totals as possible on our small glass shards while minimizing alkali ion migration and maintaining consistent analytical conditions across all measurements. While we agree that for some shards, a larger beam area would have resulted in robust measurements using the broad-beam overlap method, we prioritized consistent analytical techniques and the ability to obtain results from the largest number of glass shards possible in our samples. We will update the text to explain our justifications for the analytical conditions selected.*

On this point, I disagree with the statement made on lines 277-281 and consider that the lower Na2O values obtained relative to the published values of Cerro Hudson eruptive material is instead indicative of minor Na loss (visible in Figure 7). Consider the effect of successive analyses of particles by SEM-EDS and EPMA-WDS (inferred from Figure 2), as well as the effect of measuring multiple spots (two to three) per glass shard, the majority of which showed a decline in Na2O wt. % (see supplementary materials). It would be interesting to see the individual data, rather than averaged values, plotted against the Cerro Hudson fields to see if there is a better correlation. Presented values may also be low because of the small beam size and current used. The minor Na loss can be explained and does not adversely affect your correlation or source attribution. Please offer some consideration of these points in the text.

*We agree that the lower $Na_2O$ values could be a result of sodium migration due to successive microprobe measurements. We will update the text to clarify that while efforts were taken to minimize alkali ion migration, the lower $Na_2O$ values may be a result of beam size/current and repeated analysis, and that despite this, the majority of our measurements are still in the range of the Cerro Hudson $Na_2O$ values and thus the source attribution is not impacted by these decreased $Na_2O$ measurements. Additionally, we will provide a figure in the supplementary materials that plots the individual data points plotted with the Cerro Hudson literature data to demonstrate the correlation with the non-averaged data points.*

Minimum threshold for acceptable analytical totals: Given the literature cited (lines 154-157), please justify why analytical totals of at least 50 % were presented in this study. Without adequate justification,

it would be preferable to remove values that have analytical totals <67 % given the use of the "broad beam overlap" method of Iverson et al. (2017). However, considering the very small beam diameter used (2 µm) it may be even better to instead apply a higher minimum threshold for acceptable analytical totals (e.g., 90 %), or to discuss why lower totals were produced from shards measuring up to 15 µm in diameter.

*We agree that ideally, a higher threshold for analytical totals would provide more precise results, however, due to the characteristics of the cryptotephra analyzed, in some cases, we made the decision to include analyses with low totals and indicate to the reader which measurements had the lowest totals (i.e. below 60 wt. %). In their recent "best practices" guidelines paper, Innes et al., (2024) report good accuracy despite somewhat lower precision for EPMA analyses of very fine grained tephra shards resulting in analytical totals ranging from as low as 35 wt.% up to 101 wt. %.*

*As to why we may have returned low-totals measurements from larger glass shards, our best assessment is that the while the shards are "large" (up to 15 µm) in at least one dimension, we are only able to measure the size of the shards in two dimensions, and it is possible that the shards are very thin (either naturally, or due to the amount of polishing required), resulting in an interaction volume depth that includes the resin substrate below the shards. Additionally, due to the more complex morphology of some of the larger shards, it is also possible that the resin was not uniformly removed from the surface of the shards, resulting in some resin being included in the beam area. While this could be resolved by further polishing, due to the variable size of shards in the samples, further polishing risks fully polishing away some of the smaller shards in the sample. As a result, we chose to accept slightly lower totals in order to include analyses of as many shards as possible.*

*We will expand the text in this section to more thoroughly explain the rationale behind the analytical conditions selected, as well as the limitations of our results.*

Complete reporting of the point-by-point data: Please correct Table 1 to present the identification of 3 rhyolite shards and 1 rhyolite shard in samples 8-5 and 17-1, respectively. The analytical totals presented in the supplementary materials should also be included in Table 2, along with the exact number of analyses completed per shard for sample 14-1. This data is very important to include within the main text, particularly with regard to the consideration of possible Na-loss that may explain the discrepancies in correlation to the published data for the 1991 CE Cerro Hudson eruption visible in Figure 7.

*Thank you for pointing out the typographical errors in Table 1. They will be corrected accordingly. We have indicated in Table 2 which values correspond to low analytical totals, and will direct readers to the analytical totals for all measurements provided in the supplementary info. We will also update Table 2 to include the number of analyses conducted that comprise the averaged composition values.*

MBS-A glass morphology: The images presented in Figure 3 illustrate the sparsity of microparticles present in samples 14-1 and 17-9, however it is very difficult to see individual particles and inspect the general size and morphology of the volcanic glass shards present. Consider replacing these images, or including additional panels that present magnified images of some of the shards found in each sample.  This would better support your descriptions of glass shard morphologies in section 3.1. From the current images presented, it appears like there is more material that could have been analysed. Furthermore, it would be helpful to see images of the sparse rhyolite shards, which could be included as additional panels or in the supplementary materials.

*We will ensure the images in Figure 3 uses our best images at the highest resolution we have to demonstrate the shard morphology. Additionally, while there is a significant amount of material in the samples, EPMA analyses indicate that many of the particles in the sample are mineral grains, most likely from wind-blown terrestrial dust, rather than volcanic glass. Accordingly, we only present compositions for geochemically confirmed tephra. Additionally, while best efforts were made to section the majority of glass shards in the samples, as described previously, some of the small grains sit below the surface of the resin and were not able to be fully exposed during polishing without the risk of polishing away the already exposed shards. We will clarify in the text and figure captions that there is material present in the samples other than the glass shards conclusively identified as tephra (mineral grains, diatom fragments, etc.).*

Source eruption dynamics: The discussion of source attribution could be further improved by providing more details about the proposed source eruptions for both englacial cryptotephra horizons discovered in MBS-A. For example, what was the estimated or known duration of these recent eruptions, maximum plume heights, and total amount of ash ejected? Some of this information is presented, such as in section 4.2.1, but could be summarised earlier when first mentioning the potential source eruptions. Differences in eruption duration, style, and magnitude between the proposed source eruptions (i.e., line 313 ) could be better used to support the source attributions, complementing the geochemical correlations and atmospheric modelling presented.

*Detailed descriptions of eruption dynamics were initially left out for brevity; however we understand that they could be helpful to the reader. Published observation records exist for both the 1991 eruption of Cerro Hudson and several instances throughout the mid-1980's eruptive period of Mt Erebus. We will add text to the relevant subsections within section 4 to better describe the observed eruption dynamics of these eruptions.*

Presentation of literature values: In the supplementary materials, please include the published EPMA data that was used to create the geochemical fields and reference points presented in Figures 6, 7, and 9. Where possible include the beam diameter and analytical totals for these published major element analyses. This information would be very useful to better understand the correlative similarities and discrepancies between the published reference data and new data presented by this study, for example, the discussion of rhyolitic glass compositions in section 4.1 (lines 224-228) and similarity to shards produced by the 1991 CE Pinatubo eruption.

*We will update the supplementary materials to include a file with published EPMA data of literature values used in the figures in the manuscript.*

Minor Comments:

Title: Consider referring to the Mount Brown South ice core array, acknowledging the work undertaken on both the MBS-Main and MBS-Alpha cores.

*Thank you for the suggestion. We will consider updating the title to reflect the multiple cores used in the study.*

Line 13: Make consistent with the abstract, consider changing to "Earth system".

*We will correct this.*

Line 17: Suggest inserting a full stop after eruption histories, and begin the next sentence with "Analyzing".

*Thank you for the suggestion, we will revise these sentences.*

Line 23: Include a reference to non-sea-salt conductivity (see Winstrup et al., 2019) as another soluble tracer used to identify volcanic horizons.

*Thank you for the suggestion. This is not intended as an exhaustive list, we will consider adding further examples of soluble tracers.*

Line 24: Replace "eruptions events" with "eruption events".

*We will correct this.*

Line 26: Tephra and cryptotephra.

*We will correct this.*

Line 33: Studies reported in this paragraph mostly refer to the discovery of cryptotephra rather than tephra. The terms are used interchangeably throughout the text, creating some confusion about the nature and number of (crypto)tephra horizons discovered in Southern Ocean-Antarctic palaeoarchives.

*Throughout the manuscript, we use tephra as an umbrella term encompassing both visible and cryptotephra. We will update the text to be more specific in places in the text where confusion is likely to arise.*

Line 35: Consider a reference to the identification of (crypto)tephra in blue ice areas, which are emerging as critical archives to help further develop and refine regional tephrochronological frameworks.

*Thank you for the suggestion. We have limited our discussion here to ice core tephra, however we agree that blue ice tephra are an exciting emerging archive for tephrochronology work. We will consider adding reference to other Antarctic englacial tephra studies as well.*

Line 36: Consider replacing "eruptions" with "sources". Remove the forward slash between "Aotearoa" and "New Zealand" (here and throughout the text), and include the macron in "Taupō".

*These corrections will be made.*

Line 44: Remove "short", as there are studies cited in this sentence that investigated a range of intermediate and deep ice cores, as noted in Abbott et al. (2024). Consider including explicit reference to some of the ice cores from higher resolution sites referred to, such as WDC06A.

*Thank you for the suggestion. We will remove the specification of "short" and consider referring to specific cores in this section.*

Line 48: Consider replacing "seen in Antarctic records" with "identified in Antarctic ice core records".

*Text will be updated accordingly.*

Line 50: Replace "Aoteroa" with "Aotearoa".

*We will correct this.*

Lines 51-52: I would recommend clarifying here that only some of the volcanoes from the common source regions for volcanic products identified in Antarctic ice have been active in the satellite era. The previous sentence and cited literature refer to volcanic sources that have not been active in the satellite era or even past 1,000 years (i.e., Taupō volcano).

*We appreciate the suggestion and agree that the wording is unclear here. We will clarify that while relevant to our understanding of the types of volcanic signals found in Antarctic ice cores, some of these particular eruptions are out of the scope of the MBS shallow cores.*

Line 53: Replace "deep" with "intermediate" when referring to the MBS-Main ice core, and replace "3" with "three". Round up "25 m" to "26 m", given that the depth of the MBS-Charlie firn core extends to 25.89 m depth.

*The suggested changes will be made in the text.*

Figure 1: Please consider increasing the size of the figure and including the location of other ice cores (e.g., WAIS-Divide, RICE, Law Dome, Siple Dome, Vostok, Talos Dome, Dome Fuji) and other volcanic source regions referred to in the text.

*The final figure size will be decided upon final typesetting (two-column formatting, etc.), but we agree it should be large enough to be legible with all relevant information. We will add the locations of additional ice core sites and relevant volcanic sources to the final figure.*

Line 64: Replace "site" with "location", and include an in-text citation to support the proposed claim of "known teleconnections across the region".

*We will update the text and add references that support the teleconnections of the East Antarctic.*

Line 65: Replace "Antarctic ice cores" with "the Antarctic ice core array".

*We will correct this.*

Line 74: Mean accumulation rates?

*Yes - will be corrected.*

Line 75: How do the mean accumulation rates compare for the MBS-Bravo and MBS-Charlie firn cores?

*Thank you for raising this - in looking into this we have found an error in the text. The text will be updated to state that the mean annual accumulation rates are "0.309 ± 0.08 m yr−1 IE (ice equivalent) for the main core and 0.298 ± 0.07 m yr−1 IE for the MBS-Alpha core"*

*The accumulation rates are described in Crockart et al. (2021); the accumulation rate reported for MBS-Charlie is 0.295 ± 0.08 m yr−1 IE, and there is no published accumulation rate for MBS-Bravo. As we did not work directly with either MBS-Bravo or MBS-Charlie, however, we do not consider it necessary to include this in the text.*

Lines 81-82, and throughout: Be consistent with the use of in-text citations throughout the text, see line 80 and line 105 for contrast. Consider removing brackets around year of publication and replacing the comma with a colon, for example: "(chemistry/trace impurities: Vance et al., 2024b; Harlan et al., 2024b)".

*Thank you for pointing out the inconsistencies in referencing. We will refer to the Copernicus Publications style guide and update to ensure consistency throughout the text.*

Line 98: Fine tune.

*We will correct this.*

Lines 99-100: Good consideration, was MBS-Alpha therefore chosen out of the three surface firn cores to be used for a particular reason? Provide an example of future analyses that may require large sample volumes, such as the analysis of sulfur isotopes, iron fertilisation, or trace metals.

*We will add text to explain and justify our use of the MBS-Alpha core for this analysis as described in the reply to a similar comment made earlier in the reviewer comments above.*

Line 105: First mention of the age model, consider replacing with "on the MBS2023 chronology". Please clarify in the text whether this chronology can be applied to all four cores.

*We agree that this is an oversight - we will update the text to introduce the use of the Vance et al., (2024) MBS2023 chronology in section 2.1 of the text describing the MBS ice cores.*

Figure 2: Missing step 4. Please correct the units used for cross-sectional area (e.g., step 1 [~2 cm2] and step 5 [~10 cm2]), and replace "Alpha" with "MBS-Alpha" in the step 3 caption. Insert a hyphen between "MBS" and "Alpha" in step 5. Replace "MBS main core" with "MBS-Main ice core" in the figure caption.

*The identified errors will be corrected in the figure.*

Line 112 and throughout the text: Insert hyphen, please check that this is consistent throughout the text and figures.

*Will be corrected at this instance and throughout the text.*

Line 123: Considering the use of a small beam diameter of 2 µm for EPMA-WDS please clarify what is meant by "tephra grains of suitable size". This will will help inform why some samples were discarded for further analysis.

*This will be addressed as described earlier in this response document.*

Lines 123-125, and throughout the text: Please check the text to ensure that acronyms are defined on their first use. The acronym EPMA is first used in the Figure 2 caption and again in line 123 without being defined until line 125, whilst the acronym SEM is not currently defined in the text. Please insert "scanning electron microscopy by energy dispersive spectroscopy (SEM-EDS)" in line 125, and clarify where the FEI MLA 650 ESEM was used (see lines 143-146 for contrast).

*We will update the text to ensure acronyms are appropriately defined at first instance in the text.*

Lines 130 and 132: Please define what is meant by high(er) resolution.

*We will specify the resolution of our samples compared to the other studies referenced here.*

Lines 139-141: Consider inserting a reference to the work of Winstrup et al. (2019), which quantified the seasonality of impurity influx to Roosevelt Island visible in the RICE CFA records using the RICE17 timescale. Further comparison could be drawn between MBS and Roosevelt Island given the coastal location of both sites, and the inherent challenges faced in locating volcanic horizons.

*Thank you for the suggestion. We agree that a more thorough description of the seasonality of impurities seen in the ice core is warranted in this section. While the Winstrup et al. (2019) paper is an excellent description of seasonality of impurity influx at a coastal site, we think that it would be more appropriate in this instance to refer to Vance et al. (2024), which similarly presents the seasonality of impurities, as it is specific to the MBS site, and describes some of the complexities of MBS.*

*It is a good suggestion to refer to the Winstrup et al. (2019) Roosevelt Island work as comparison when it comes to coastal Antarctic ice cores, and we will consider incorporating reference to this into the discussion section.*

Line 145: Five WDS spectrometers (see Table 2 caption for comparison).

*We will correct this.*

Lines 147-148: Please state here which elements were analysed.

*The text will be updated to specify the elements analyzed.*

Line 149: Provide a reference for the secondary glass standards used.

*Reference for secondary standards will be provided.*

Lines 152-153: Please revise this statement, as 27 of the 52 EPMA-WDS analyses presented in the supplementary materials have analytical totals <90 %.

*We will update the text to be more specific when referring to how many samples have totals over 90%.*

Line 158: Replace "based on the as recommended" with "as recommended".

*We will correct this.*

Line 161: HYSPLIT acronym first used in section 2.2 on line 92.

*Will update the text to define the HYSPLIT acronym at first instance.*

Line 170: BSE.

*We will correct this.*

Line 174: Consider referring to the sample ID rather than sample depths, for example: "Glass shards were most abundant in sample 14-1 (13.28-13.34 m depth)".

*Thank you for the suggestion. We made the deliberate decision to refer to the samples only by depth and/or age, as the sample IDs are not useful to the reader without knowledge of their depth, and are explained in Table 1.*

Line 175, and throughout the text: Be consistent with units, replace micron(s) with μm.

*We will correct this for consistency throughout.*

Lines 176-178: Please clarify as it is uncertain from the current wording if only one of the glass shards identified in this sample was found to be >10 μm in diameter (in this case ~15 μm).

*You are correct that one shard was larger than 10 μm (~15 μm). The wording in the text will be updated to clarify this.*

Table 1 Caption: Please clarify in the caption that this is a "Summary of the 12 MBS-Alpha core samples analyzed by EPMA-WDS", as a total of 48 of the 70 prepared samples were reported to contain volcanic glass shards. Here and throughout the text, please be consistent with the chosen expanded abbreviation for TAS (contrast presentation in the caption for Table 1 with that of Figures 4, 6, 7, and 9, as well as line 183).

*Figure and table captions will be updated as needed.*

Line 185: Consider reordering in order of particle abundance.

*We will correct the text accordingly.*

Line 186: One andesite.

*We will correct this.*

Line 187, and throughout the text: Ensure capitalisation for in-text references to the Tables, as with Figures.

*Will update to ensure consistent capitalization throughout the text.*

Line 190, and throughout the text: Please be consistent with reference to tables and figures, contrast with line 197 (Fig. vs. Figure).

*The text will be corrected to ensure that it follows the Copernicus style guide for figure and table referencing.*

Line 199: Trace element chemistry?

*We are referring here to trace elements as well as trace ions.*

Line 204: Replace "non-seasalt sulfate" with "non-sea-salt sulfate (nssSO4-2)", and revise accordingly in the text caption for Figure 5 and line 245.

*The text will be updated accordingly.*

Table 2: In the caption, replace "glass tephra shards" with "volcanic glass shards". Please consider using another symbol for particle 18-1_014.

*The table caption will be updated accordingly.*

Line 212: Majority? Only 12 samples analysed by EPMA-WDS of the initial 70. Does this refer to data from SEM-EDS not presented in this study?

*The text will be updated to specify that here we refer to "the majority of tephra-bearing sample depth ranges."*

Figure 4: Why not present sample ID rather than mean depths? The symbol used here for sample 18_1 is different to Table 2 but much easier to read.

*We have chosen to refer to sample mid-depth throughout as the sample ID numbering system does not directly relate to depth, and thus does not provide meaningful information about the sample to the reader.*

Figure 5 Caption: Water isotopes.

*Figure caption will be updated accordingly.*

Lines 242-243: Interesting discussion in section 4.1, which could be better supported by providing images of some of the rhyolite glass analysed by EPMA-WDS to illustrate their size and morphology in comparison to the proposed primary dacite and phonolite populations.

*Where possible, we will include BSE images of additional samples in the supplementary materials.*

Line 247: Delete additional space in "sample , we".

*We will correct this.*

Line 257: What literature values? Please provide a reference.

*References will be provided earlier in the sentence to clarify the literature values used.*

Line 271: Remove "13.3" and replace with "1991".

*We will correct this.*

Line 274: Reported literature values.

*We will correct this.*

Line 295: Please revise the Cerro Hudson eruption dates to "August 8-15, 1991".

*Date will be corrected.*

Lines 328-331: I would also recommend referring to the age of the ice that the dacite cryptotephra was found within relative to the nssSO42- peak of Pinatubo (i.e., the MBS2023 chronology and trace impurities from Figure 5). These are convincing lines of supporting evidence for your proposed source attribution.

*We have chosen not to rely on this line of evidence, as the two-month period between the eruptions of Pinatubo (June) and Cerro Hudson (August) is below our ability to resolve based on dating uncertainty and the ~5 cm samples used for this work. We will update the manuscript to include this reasoning.*

Line 333: Replace "volcanoes" with "volcanic eruptions".

*This will be corrected in the text.*

Figure 9: It is difficult to read and distinguish some of the published values from the presented values. Please consider including the cited data used here within a supplementary file.

*As indicated previously, the supplementary data files will be updated to include literature values where possible.*

Line 374: The 1985 cryptotephra, please be consistent throughout (see line 376).

*The text will be corrected to ensure consistent phrasing.*

Lines 375-377: Excellent point, but it is likely that the shards are too small to analyse for trace element compositions by LA-ICP-MS.

*The shards may be small for trace element analysis by LA-ICP-MS, but we will retain this point in the text as other techniques (possibly synchrotron XFM?) or future technological developments may make these measurements possible.*

Line 411: Such as? Please state what eruptions you are referring to here - are they the eruptions presented in line 432?

*Here we refer to Table 3 in Vance et al., 2024. We will update the text to reflect this.*

Line 416: MBS as a cryptotephra archive.

*We will correct this.*

Section 5.3. There are some intriguing claims and findings proposed, however I find this section of the discussion difficult to follow as it lacks the clarity and structure of previous sections.

*We appreciate the feedback. We will carefully review and check section 5.3 for clarity.*

Line 480: 42 volcanic glass shards.

*Will update the text to be more specific here.*

Line 481: Please revise this sentence as only 13 shards were reported from sample 14-1 (1991 CE) and this is not an entirely homogenous population (with 12 dacite and 1 andesite shards analysed).

*Text will be updated to correct the typo and clarify that the populations are "largely homogeneous."*

Reference List: Please check each reference to ensure that no key elements are missing, such as author names, the DOI, page numbers, volume number, and journal titles. There are inconsistencies and minor errors throughout the reference list that need to be addressed.

*The reference list will be reviewed and corrections will be made as needed.*

Supplementary Materials, MBS-Alpha Glass Worksheet: Please revise the caption to reflect that the analyses were completed on samples from the MBS-Alpha core. Apply subscript for element titles, align the column headings with the data presented below, and please make the cited reference consistent with the presentation and format used in the main text. Is there a data point missing for sample 14-1_025a?

*Corrections to the spreadsheet will be made as needed.*

Supplementary Materials, Secondary Standards Worksheet: Could you please clarify which MBS-A EPMA datasets the three analytical sessions and therefore secondary standards data presented align with.

*Supplementary data will be updated to include the analytical sessions corresponding with secondary standards data.*

*References:*

*Crockart, C. K., Vance, T. R., Fraser, A. D., Abram, N. J., Criscitiello, A. S., Curran, M. A. J., Favier, V., Gallant, A. J. E., Kittel, C., Kjær, H. A., Klekociuk, A. R., Jong, L. M., Moy, A. D., Plummer, C. T., Vallelonga, P. T., Wille, J., and Zhang, L.: El Niño–Southern Oscillation signal in a new East Antarctic ice core, Mount Brown South, Climate of the Past, 17, 1795–1818, https://doi.org/10.5194/cp-17-1795-2021, 2021.*

*Innes, H. M., W. Hutchison, and A. Burke.: Geochemical analysis of extremely fine-grained cryptotephra: New developments and recommended practices, Quaternary Geochronology 83, page 101553. https://doi.org/10.1016/j.quageo.2024.101553, 2024.*

*Vance, T. R., Abram, N. J., Criscitiello, A. S., Crockart, C. K., DeCampo, A., Favier, V., Gkinis, V., Harlan, M., Jackson, S. L., Kjær, H. A., Long, C. A., Nation, M. K., Plummer, C. T., Segato, D., Spolaor, A., and Vallelonga, P. T.: An annually resolved chronology for the Mount Brown South ice cores, East Antarctica, Climate of the Past, 20, 969–990, https://doi.org/10.5194/cp-20-969-2024, 2024.*

*Winstrup, M., Vallelonga, P., Kjær, H. A., Fudge, T. J., Lee, J. E., Riis, M. H., Edwards, R., Bertler, N. A. N., Blunier, T., Brook, E. J., Buizert, C., Ciobanu, G., Conway, H., Dahl-Jensen, D., Ellis, A., Emanuelsson, B. D., Hindmarsh, R. C. A., Keller, E. D., Kurbatov, A. v., … Wheatley, S.: A 2700-year annual timescale and accumulation history for an ice core from Roosevelt Island, West Antarctica, Climate of the Past, 15(2), 751–779. https://doi.org/10.5194/cp-15-751-2019, 2019.*